# Thermobarometry and Geochemistry of Mantle Xenoliths from Zapolyarnaya Pipe, Upper Muna Field, Yakutia: Implications for Mantle Layering, Interaction with Plume Melts and Diamond Grade

**Igor Ashchepkov [1],***, **Nikolay Medvedev [2]**, **Nikolay Vladykin [3]**, **Alexander Ivanov [4]** and **Hilary Downes [5]**

[1] Sobolev V.S. Institute of Geology and Mineralogy SB RAS, 630090 Novosibirsk, Russia
[2] Nikolaev Institute of Inorganic Chemistry SB RAS, 630090 Novosibirsk, Russia; medvedev@niic.nsc.ru
[3] Vinogradov Institute of Geochemistry SD RAS, 664033 Irkutsk, Russia; vlad@igc.irk.ru
[4] Central Science and Research Geology and Prospecting Institute of the Stock Company "ALROSA",
    678170 Mirny, Russia; ivanovas@alrosa.ru
[5] Department of Earth and Planetary Sciences, Birkbeck College, University of London,
    London WC1E 7HX, UK; h.downes@ucl.ac.uk
*** Correspondence: igor.ashchepkov@igm.nsc.ru; Tel.: +79-139-872-605

**Abstract:** Minerals from mantle xenoliths in the Zapolyarnaya pipe in the Upper Muna field, Russia and from mineral separates from other large diamondiferous kimberlite pipes in this field (Deimos, Novinka and Komsomolskaya-Magnitnaya) were studied with EPMA and LA-ICP-MS. All pipes contain very high proportions of sub-calcic garnets. Zapolyarnaya contains mainly dunitic xenoliths with veinlets of garnets, phlogopites and Fe-rich pyroxenes similar in composition to those from sheared peridotites. PT estimates for the clinopyroxenes trace the convective inflection of the geotherm (40–45 mW·m$^{-2}$) to 8 GPa, inflected at 6 GPa and overlapping with PT estimates for ilmenites derived from protokimberlites. The Upper Muna mantle lithosphere includes dunite channels from 8 to 2 GPa, which were favorable for melt movement. The primary layering deduced from the fluctuations of CaO in garnets was smoothed by the refertilization events, which formed additional pyroxenes. Clinopyroxenes from the Novinka and Komsomolskaya-Magnitnaya pipes show a more linear geotherm and three branches in the P-Fe# plot from the lithosphere base to the Moho, suggesting several episodes of pervasive melt percolation. Clinopyroxenes from Zapolyarnaya are divided into four groups according to thermobarometry and trace element patterns, which show a stepwise increase of REE and incompatible elements. Lower pressure groups including dunitic garnets have elevated REE with peaks in Rb, Th, Nb, Sr, Zr, and U, suggesting mixing of the parental protokimberlitic melts with partially melted metasomatic veins of ancient subduction origin. At least two stages of melt percolation formed the inclined PT paths: (1) an ancient garnet semi-advective geotherm (35–45 mW·m$^{-2}$) formed by volatile-rich melts during the major late Archean event of lithosphere growth; and (2) a hotter megacrystic PT path (Cpx-Ilm) formed by feeding systems for kimberlite eruptions (40–45 mW·m$^{-2}$). Ilmenite PT estimates trace three separate PT trajectories, suggesting a multistage process associated with metasomatism and formation of the Cpx-Phl veinlets in dunites. Heating associated with intrusions of protokimberlite caused reactivation of the mantle metasomatites rich in $H_2O$ and alkali metals and possibly favored the growth of large megacrystalline diamonds.

**Keywords:** kimberlite; mantle; upper muna; pyrope garnet; clinopyroxene ilmenite; thermobarometry; geochemistry

## 1. Introduction

The Upper Muna kimberlite field in Yakutia (Figure 1) has been involved in diamond exploration since 2018. Other large kimberlite pipes in the central part of Yakutian kimberlite province (YKP) [1] were exhausted and excavation in open quarries stopped except for Yubileinaya [2], Zarnitsa, Komsomolskaya and Zarya pipes [3] in the Alakit field. The largest pipes are now explored by underground mining in the Daldyn, Alakit and Malo-Botuobinsky regions [4]. There are difficulties in finding productive commercial kimberlites that may be easily exploited.

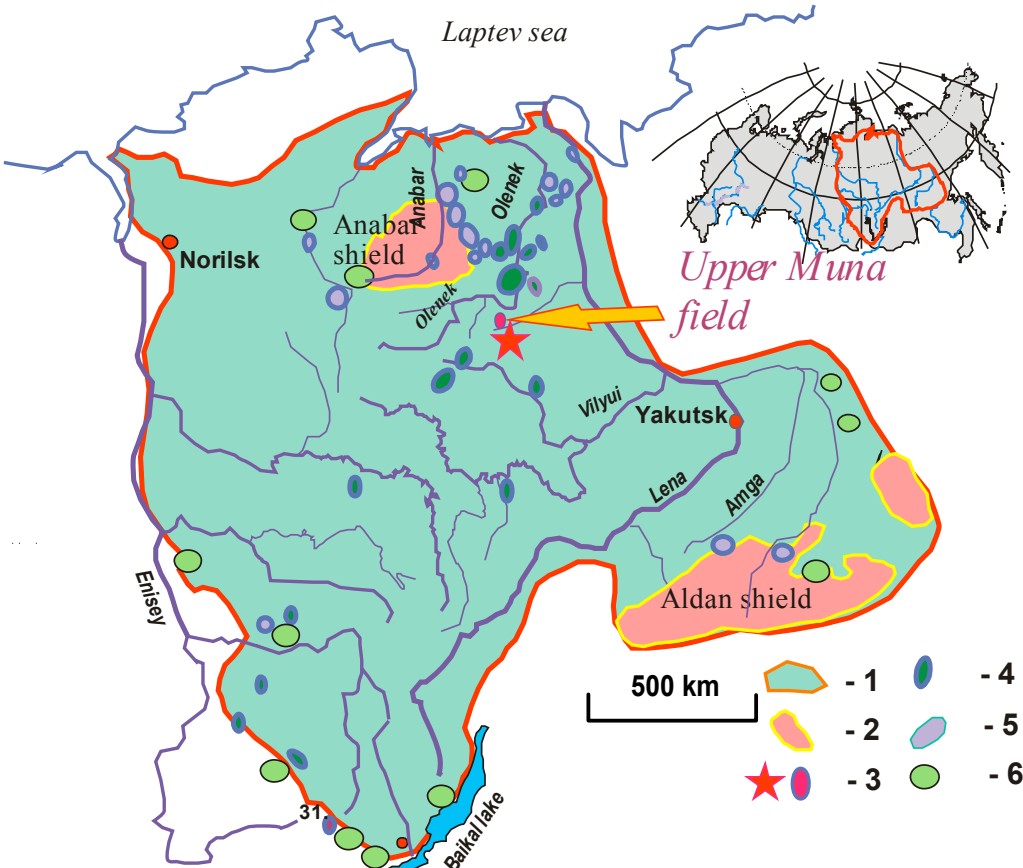

**Figure 1.** Location of the Upper Muna field and other kimberlite fields in Siberian platform 1. Siberian platform. 2. Shields. Kimberlite fields: 3. Upper Muna Field; 4. Late Devonian fields; 5. Lower Triassic and Jurassic fields [4,5]; 6. Carbonatite massifs.

The new prospective regions of the Upper Muna kimberlite field are located 170 km NNE from the Udachny working settlement and comprise more than 20 kimberlite bodies (Figure 2) [5] with several relatively large pipes such as Zapolyarnaya, Deimos, Komsomolskaya-Magnitnaya, and Novinka (Figure 3a). Two quarries were planned to explore these pipes. Excavation of the Zapolyarnaya pipe started in autumn 2018 and yielded many large gem-quality diamonds (average price $210 per crt) up to 98.8 crt (Figure 3b,c). The largest diamond found was 256 crt. Some crystals have yellow (Figure 3d,e) and brownish tones (Figure 3f). The diamond grade should allow production of 2 million crt per year for more than 25 years. Zapolyarnaya contains most of the high-quality gem diamonds in the Upper Muna cluster and now this pipe is being extensively mined (Interfax, Reports from ALROSA Company [6]). The large well-shaped diamonds are close to octahedral, whereas the smaller crystals are often rounded. These diamonds are mainly rhombdodododecahedra or dodecahedra and different combinations of forms and, more rarely, octahedral with rounded shape and signs of dissolution [7]. Diamonds from Zapolyarnaya are characterized by low N contents and light $\delta^{13}C$ and are suggested to have originated from fluids [8].

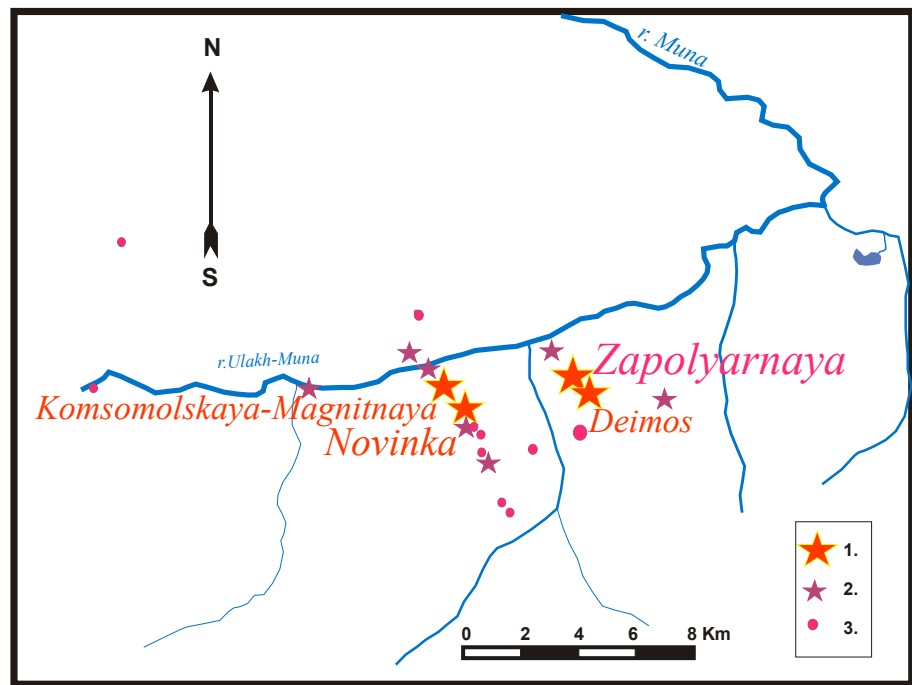

**Figure 2.** Scheme of the kimberlite pipes' location in Upper Muna field [5]. 1. Large pipe excavating in quarries. 2. Large pipes without excavations. 3. Small kimberlite pipes.

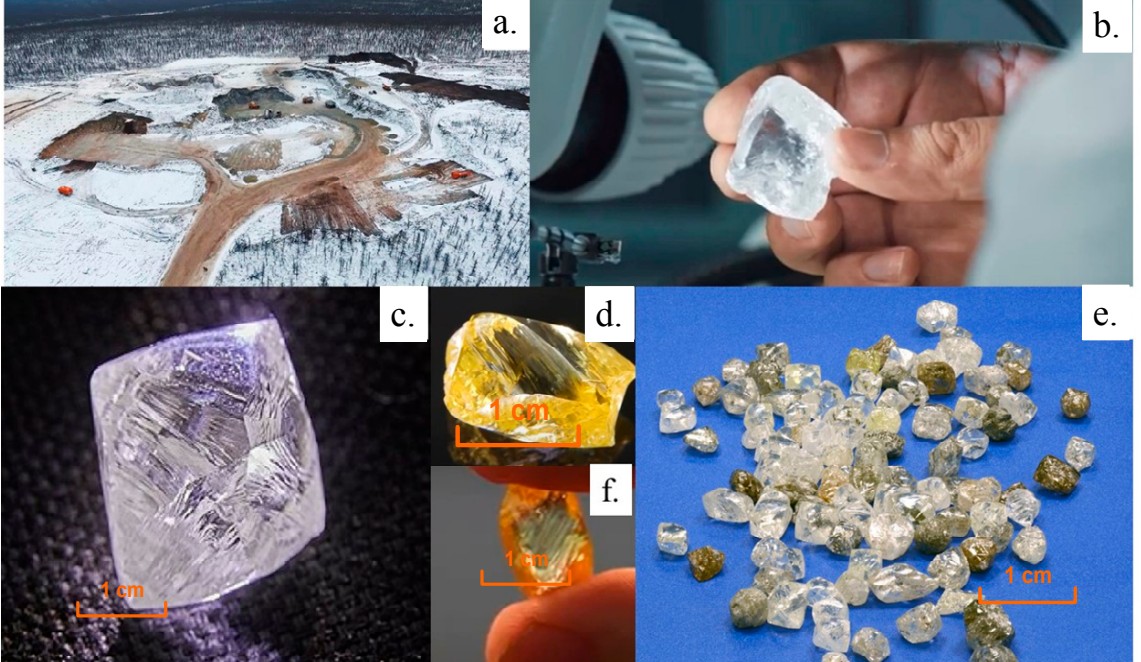

**Figure 3.** (**a**). Photo of the open pit in the Zapolyarnaya pipe Upper Muna field. (**b**). Large gem quality diamond from Zapolyarnaya pipe. (**c**–**f**)-diamonds from Zapolyarnaya pipe.

The field was described very briefly in early Russian publications [1,2,9,10]. General features of the kimberlite indicator minerals (KIM) were also described [10–13]. The features of kimberlite indicator minerals (KIM) in concentrates from Upper Muna differ slightly from those of kimberlites from the Central Yakutian provinces, and the diamond inclusions also have some specific features.

Preliminary mineralogical information about Upper Muna pipes was presented in reports to ALROSA and monographs devoted to kimberlite indicator minerals (KIM) of the whole

YKP [9,10]. Clinopyroxene major elements from Novinka pipe were published by Ziberna et al. [14], yielding a pyroxene geotherm for clinopyroxene (Cpx) concentrate. Mantle xenoliths from Novinka pipes [14] showed the 40 mW·m$^{-2}$ geotherm for clinopyroxenes from mantle xenoliths from the Komsomolskaya-Magnitnaya [15] pipe with divisions in pressure into several groups, suggesting mantle layering.

The purposes of this study are to record more complete information about the composition and structure of the richest diamond pipe in the Upper Muna field using mantle xenoliths and concentrates, and to use these data to distinguish mantle evolution processes and their influence on diamond grade.

We present compositions of clinopyroxenes (Cpx), garnets (Gar), chromites (Chr), ilmenites (Ilm), amphiboles (Amph), and phlogopites (Phl) and use them to reconstruct the mantle structures beneath the Upper Muna field, using both mantle xenoliths and concentrates from four kimberlite pipes. We describe the geochemistry of a large number of minerals from xenoliths and concentrates obtained by EPMA and LA-ICP-MS.

All these data show that the mantle column beneath the Zapolyarnaya pipe was extremely depleted and metasomatized in ancient times. We suggest that protokimberlite magmas were responsible for the heating and refertilization of the mantle column and possibly were responsible for the growth of large diamonds.

## 2. Geology and General Information

About 20 kimberlite pipes and dykes are located in the Upper Muna valley and its shoulders at the intersection of a large sub-lateral fault with two NNW faults located 5 km apart (Figure 2) [5]. All of them are Devonian according to perovskite U-Pb [16] and earlier Rb-Sr and Sm-Nd dating for kimberlite minerals [5]. The geology and composition of kimberlites, and the general features of kimberlites and KIM were described previously [4,10,13]. Macrocrystic kimberlites and autholitic breccias show high MgO (32–30 wt.%) and low $Al_2O_3$ (1.76–2.37 wt.%) contents [4,13] and a relatively low carbonate component. The most Mg-rich kimberlite is from the Zapolyarnaya pipe. They are all Group I kimberlites tending to the high field HIMU (high $^{238}U/^{204}Pb$) field in the $^{87}Sr/^{86}Sr$-$\varepsilon$Nd diagram [4,13]. The $^{87}Sr/^{86}Sr$ ratios correlate positively with $CO_2$ content. Trace element geochemistry was presented in comparison with kimberlites from other fields [4]. Compared with other pipes, those from Upper Muna have slightly higher (La/Yb)$_n$ (normalized to [17]) ratios close to those determined for Yubileinaya [3,4] and slightly higher levels of large ion lithophile elements (LILE) and Sr, suggesting relatively low degrees of melting and participation of hydrous fluids during melting.

The kimberlites contain all the common KIM typical of kimberlites in Yakutia and other provinces: garnets (Gar), clinopyroxene (Cpx), orthopyroxene (Opx), chromites (Chr), ilmenite (Ilm), and amphiboles (Amph). However, Upper Muna shows high concentration of sub-Ca garnets of both harzburgitic and dunitic types (G9-G10), according to the international classification [18,19]. They cover practically the entire interval from 1.5 to 14 wt.% $Cr_2O_3$. Clinopyroxenes are found in concentrates of all pipes and are most abundant in Novinka [14] and Komsomolskaya-Magnitnaya [15] pipes and rarer in Zapolyarnaya. Amphiboles are more common in Deimos pipe and compile ~11% relative to Cpx, although they are present in all others bodies. Ilmenites are more common in Deimos and Novinka (up to 10% from concentrate) pipes and chromites prevail in Zapolyarnaya and Poiskovaya pipes (7% of the total concentrate). Opx grains rarely occur in concentrate.

## 3. Samples

The xenoliths were collected from the intermediate kimberlite store in Udachny. More than 200 xenoliths were collected, but only 52 of them contain fresh Cpx and only 3 Opx (Figure 4a–d). Phlogopites are frequent whereas Cr-amphiboles occur more rarely. Xenoliths from Zapolyarnaya are mostly harzburgites and dunites with less abundant lherzolites, which seem to be slightly refertilized with scattered garnet pyroxenite veinlets with phlogopites. They vary widely in grain size and many varieties contain fine-grained granular "sugar-like" olivine grains (0.1–1 mm in dimensions) similar to

those found in deformed peridotites of Udachnaya [18]. Harzburgites and nearly pure dunites contain rare garnets as porphyroclasts or as small disseminated grains. Dark serpentine clusters after Opx are rarely surrounded by Cpx, but most xenoliths contain fresh olivine grains. The previously described giant-grained garnets in dunites [19] were not found. Cr-bearing pyroxenites are extremely rare.

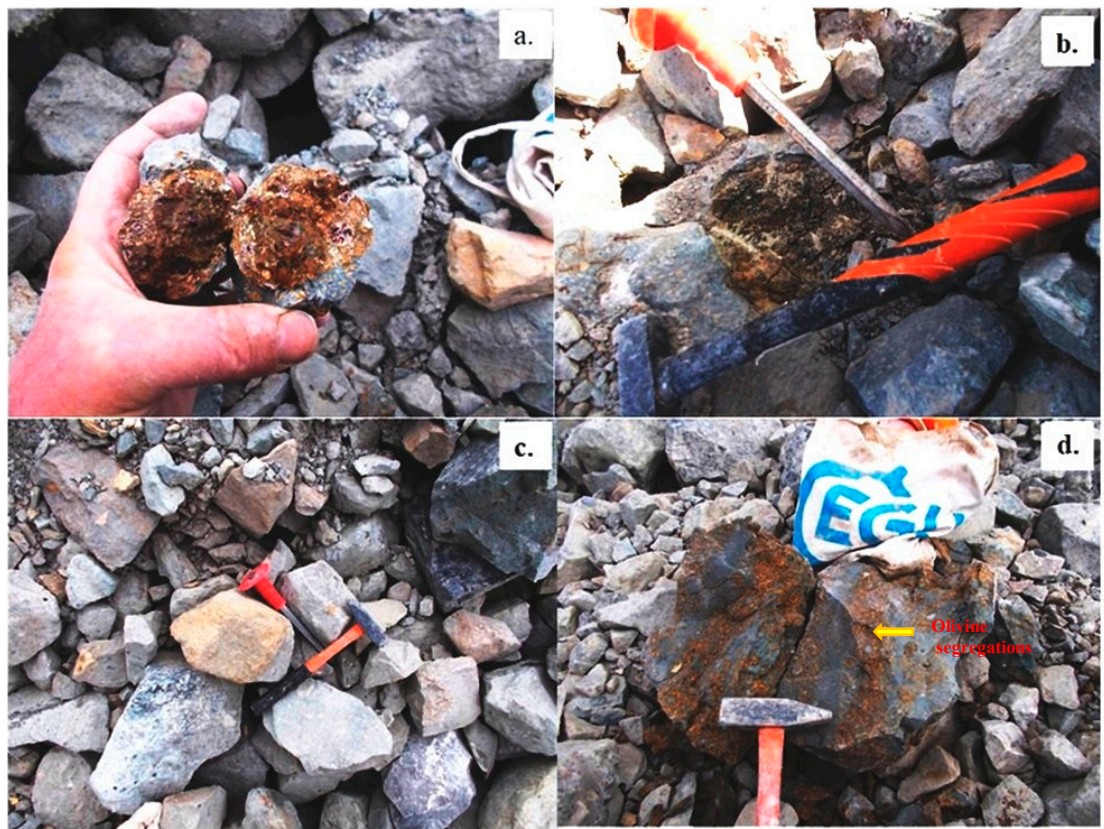

**Figure 4.** Photo of the samples from Zapolyarnaya pipe. (**a**,**b**) Mantle xenoliths from Zapolyarnaya pipe with visible Cr-diopsides and garnets. (**c**) Large mantle xenolith from Zapolyarnaya pipe. (**d**) Segregations of the olivine grains and dunite debris in kimberlites. The brownish aggregates are altered olivines, the light grains are fresh olivines, dark grey is kimberlite.

Pale olivine aggregates are frequent in Zapolyarnaya kimberlites and resemble those found in polymict breccias [20] or essentially olivine mush consisting of disintegrated mantle rocks in kimberlites [21] or form light polycrystalline mosaic light "sugar-like" olivine (+Gar, Ilm, Sp) aggregates similar to those described in sheared peridotites [15,18] in the darker kimberlites. They occur as tubes or channels (Figure 4d). We included some such ilmenites and chromites from these segregations to obtain more complete geotherms and P-T-X diagrams. Heavy mineral concentrates from the large diamondiferous pipes were obtained from crushed drill cores from the deeper horizons with fresh olivine using washing and magnetic separation methods in the Vinogradov Institute of Geochemistry SB RAS. They were then mounted in epoxy resin and polished.

## 4. Methods

Preliminary electron microprobe analyses (EPMA) of mineral compositions were determined using a Camebax Micro electron microprobe in Sobolev V.S. Institute of Geology and Mineralogy of Siberian division Russian Academy of Sciences (IGM SB RAS), and trace element (TRE) analyses were made using inductively coupled plasma mass spectrometry with laser ablation (LA-ICP-MS) method on mass-spectrometer Finnigan Element for Zapolyarnaya and Novinka pipes in IGM SB RAS [11],

which is used also in the article, and we made a new series of analyses with new equipment in Nikolaev Institute of Inorganic Chemistry SB RAS (NIIC SB RAS).

### 4.1. Electron Microprobe Analyses (EMPA)

Xenoliths were crushed, and selected mineral grains were analyzed in Analytic Center of IGM SB RAS. Electron microprobe analysis of studied minerals indicator kimberlite minerals (KIM) (garnets, Cr-diopsides, augites, micas, ilmenites and chromites) was performed using Camebax Micro (xenoliths) and Jeol JXA8320 according to the established procedure [22]. Beam was focused to 1 μm. The accelerating voltage was 15 kV, and the beam current was 15 nA with 15 s counting time. Relative standard deviation did not exceed 1.5%; the precision was close to 2–5% 2 sigma error. Natural minerals and synthetic materials were used as the standards. The natural garnet U-92 was used for control and accuracy. We analyzed one spot using 10 repetitions on grain without beam movement. The EPMA data are in the Table S1.

### 4.2. Inductively Coupled Plasma Mass Spectrometry with Laser Ablation (LA-ICP-MS)

About 20 garnet and Cpx pairs from Zapolyarnaya peridotitic xenoliths (lherzolites, harzburgites, dunites) and KIM from Zapolyarnaya Gar (18) and Cpx (9) and Novinka Gar (25) and Cpx (13), micas (2) and amphiboles (1) were analyzed by inductively coupled plasma mass spectrometry with laser ablation (LA-ICP-MS) in NIIC SB RAS using an iCAP Q mass spectrometer (Thermo Scientific, Waltham, MA, USA) and a NWR 213 (New Wave Research), Nd YAG: UV 133 nm laser ablation system. This is a 213 nm UV laser ablation system based on a frequency-quintupled 1064 nm IR solid state (analyst N.S. Medvedev). The method has a detection limit ~$10^{-7}$ wt.% (~$10^{-3}$ ppm) and standard deviation of the measurements for most isotopes was about 8–25%. In total, 58 isotopes of elements were analyzed. The NIST 612, 610 SRM were used as the standards. For the internal control, $^{24}$Mg, $^{29}$Si, $^{39}$K, $^{47}$Ti, $^{55}$Mn, $^{52}$Cr, and $^{40}$Ca isotopes were used. The agreement with EPMA analyses was checked and the trace element level was estimated. The isotope $^{40}$Ca was used as the internal standard. Additionally, garnets and clinopyroxenes from samples 315–254 and 315–73 were dissolved and analyzed by ICP-MS [23] and used as internal standards to check the agreement of REE and spider diagram patterns. The agreement of the control grains of LP-ICP-MS analysis and ICP-MS analysis of the sample solutions analyses are visible in Table S2. Analyses were made in two long series and there are no systematic differences between the element concentrations. All data are presented in tables of the Tables S1 and S2.

## 5. Mineralogy

*Garnets* from the Upper Muna kimberlites plot in the lower part of the lherzolite field up to 14 wt.% $Cr_2O_3$ in the $Cr_2O_3$-CaO diagram, but most have harzburgitic and dunitic compositions and pyroxenitic types are very rare. The most representative and continuous garnet trend for Zapolyarnaya pipe reveals the rarity of pyroxenitic varieties (Figure 5a). Sub-calcic garnets, belonging to G9 according to [24,25], appear from 1.8 wt.% $Cr_2O_3$, forming sub-trends parallel to the $Cr_2O_3$ axis. Enrichment to 1.2 wt.% $TiO_2$ is more visible in the left part of $Cr_2O_3$ axis; the $Na_2O$ elevation to 0.3 wt.% in Zapolayrnaya garnets is lower than in other pipes in Upper Muna field.

In the Novinka pipe, the dunite-harzburgite garnets are restricted to 5–12 wt.% $Cr_2O_3$ and in the Deimos pipe from 2 to 11 wt.% (Figure 5b). The dunitic ones continue the trend to 14 wt.% $Cr_2O_3$. Most of them fall at the upper boundary of the harzburgites field but dunite garnets also appear in the 5–11 wt.% $Cr_2O_3$ interval. Garnets from Komsomolskaya-Magnitnaya and Deimos pipes (Figure 5c,d) contain from 2 to 12 wt.% $Cr_2O_3$. Evidence for Ti, Na metasomatism of garnets is more pronounced in Novinka and Komsomolskaya-Magnitnaya pipes.

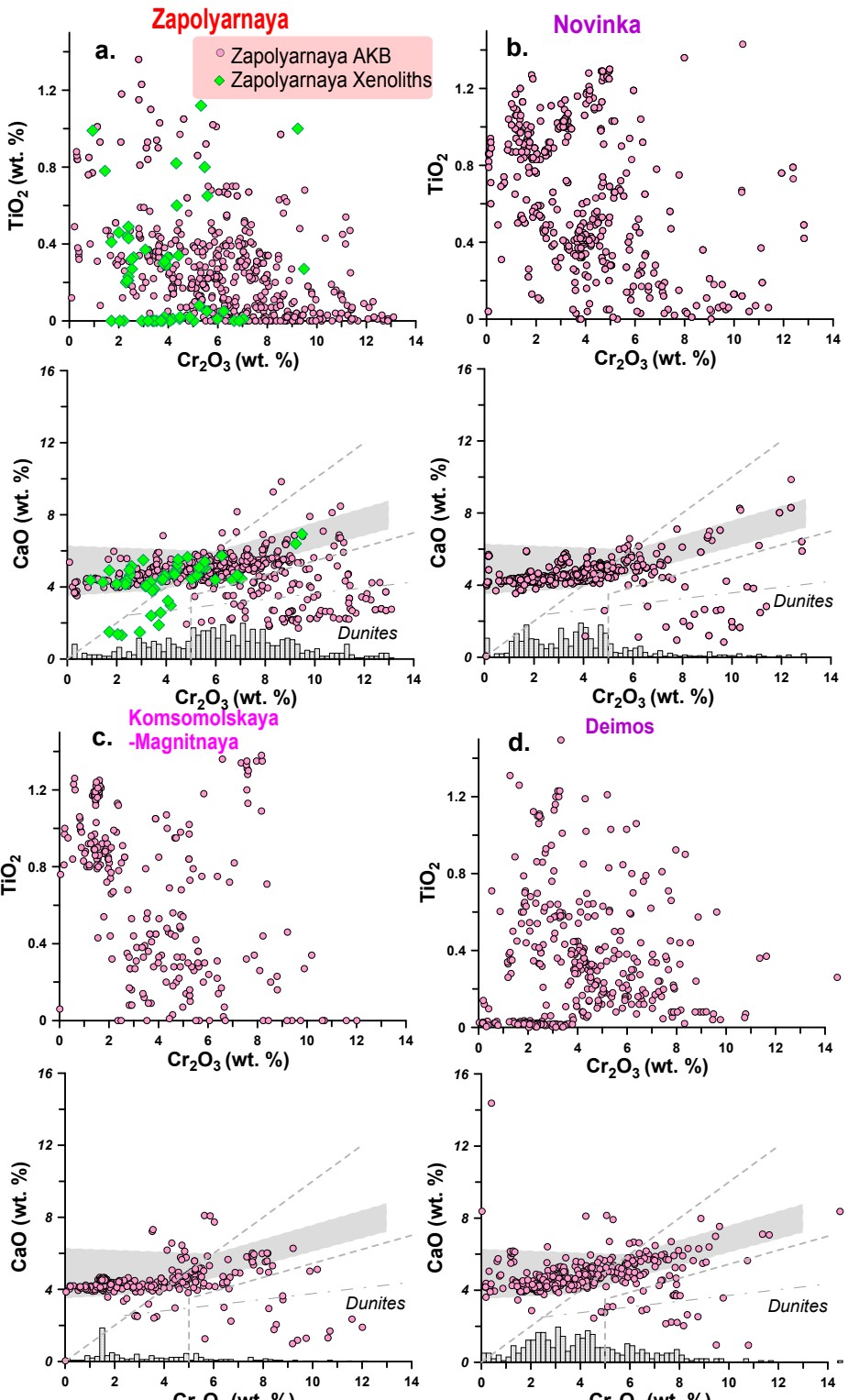

**Figure 5.** Compositions of Cr-garnets from Upper Muna (xenolith and concentrate) (**a**) Zapolyarnaya; (**b**) Novinka; (**c**) Deimos, (**d**) Komsomolskaya-Magnitnaya. Data for Novinka, Deimos, and Komsomolskaya-Magnitnaya are partly published [12]. Grey field for lherzolites. The dashed line is for diamond association [1]. The dotted-dashed line divides dunite associations. The histograms show wt.% $Cr_2O_3$ and use the same axes. Data are from this study and from previous publications [11,12].

*Clinopyroxenes* from the xenoliths of the Zapolyarnaya pipe show a variation in FeO from 1 to 6 wt.% and have nearly constant $Al_2O_3$ around 1.5–2.5 wt.% (Figure 6a). Only a few Cpx close to omphacites

have 6–10% wt.% $Al_2O_3$ (Figure 6a). Enrichment in $Cr_2O_3$ to 5 wt.% of the Fe-poor varieties (to 3 wt.% FeO) falls to 2 wt.% in those with 6 wt.% FeO. In general, pyroxenes with 3.2–4.5 wt.% FeO are similar to those from deformed peridotites. More Fe-rich varieties are similar to the Ilm-bearing associations and intergrowths with ilmenites like those found in Dalnyaya pipe [26]. However, $TiO_2$ content is low (to 0.4 wt. %), increasing for Fe-rich varieties, tending to Al-augite compositions. Cpx are relatively common in the concentrate from all other Upper Muna pipes (Figure 6b). In general, they show similar trends as for Zapolyarnaya pipe, but variations in $TiO_2$ and $Al_2O_3$ are wider. Novinka Cpx show much wider variations in $Al_2O_3$ up to 6 wt.% in the Fe-poor part of the trend (Figure 6b). $TiO_2$ increases together with FeO to 0.5 wt. %. Cpx from Deimos pipe show the strongest Cr-Al variations, probably due to metasomatism and the presence of eclogite types of Cpx in concentrate (Figure 6b).

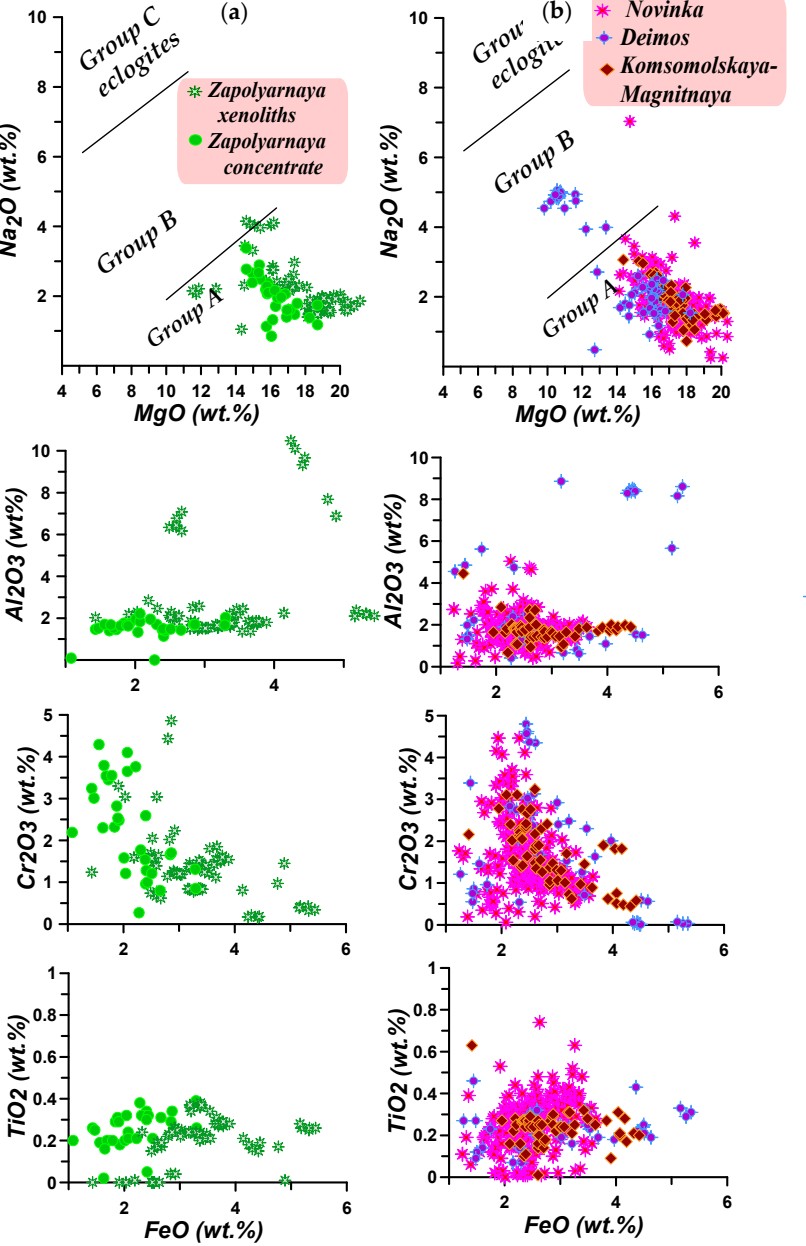

**Figure 6.** Compositions of Cr-diopsides, augites and omphacites from Upper Muna field. (**a**) Zapolyarnaya; (**b**) Novinka; Deimos, Komsomolskaya-Magnitnaya. Data for Novinka, Deimos, and Komsomolskaya-Magnitnaya are partly published [12]. Concentrate means heavy mineral separates.

*Ilmenites* from Zapolyarnaya (Figure 7a) show wide variations in $Cr_2O_3$ content with enrichment up to 6 wt.% for the richest $TiO_2$ varieties: 56 to 50 wt.% The $Al_2O_3$ and NiO abundances are also higher than for low $TiO_2$ ilmenites, and are less contaminated in $Cr_2O_3$, $Al_2O_3$ and NiO derived from peridotites. In Deimos pipe, ilmenites are rather abundant and show variations typical for those from kimberlites of Alakit and Daldyn region with nearly linear $FeO$-$MgO$-$TiO_2$ relations (Figure 7b) [3,11,26,27], usually considered to be a fractionation trend.

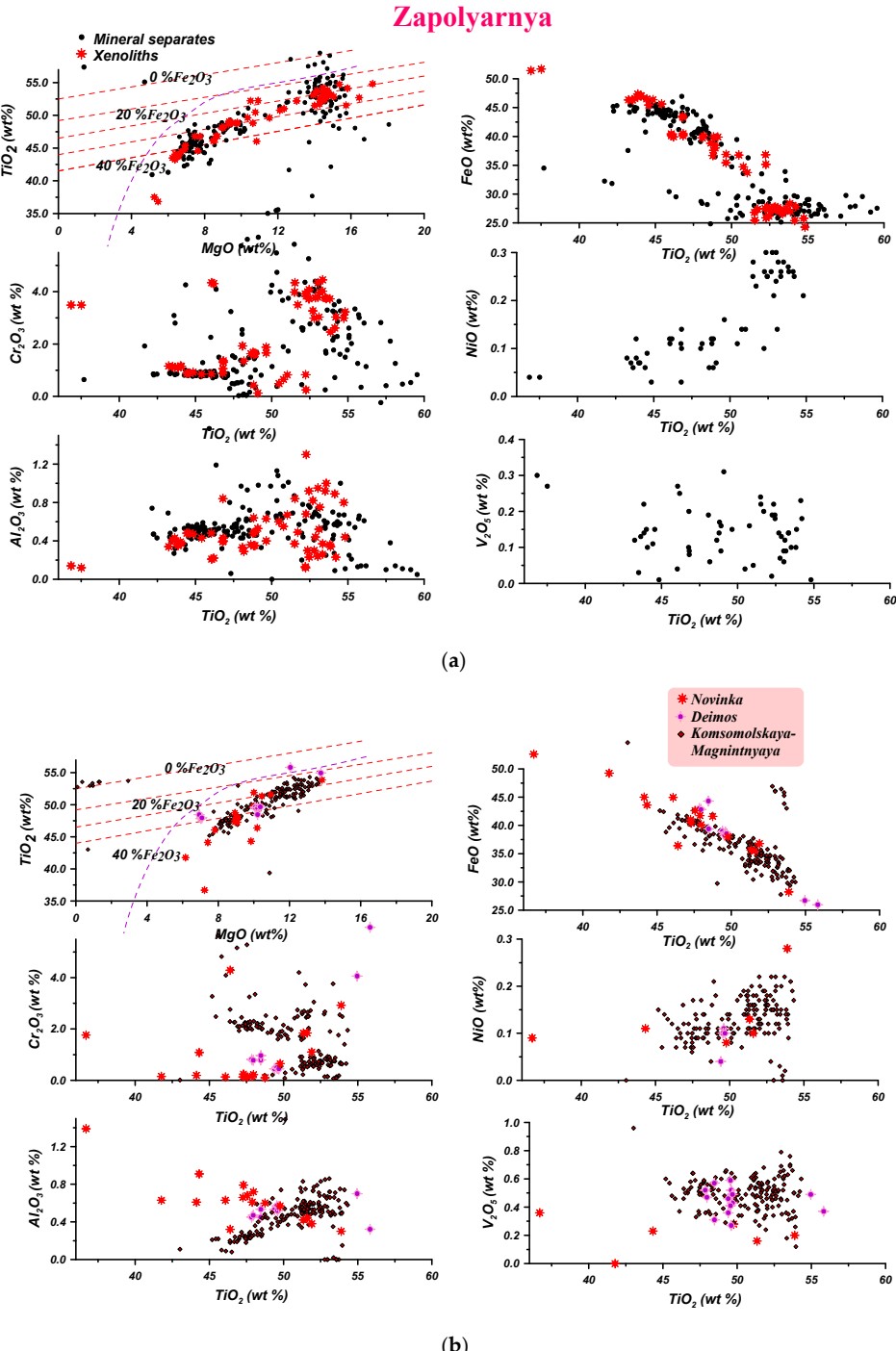

**Figure 7.** Compositions of ilmenites from Upper Muna field. (**a**) Zapolyarnaya; (**b**) Novinka; Deimos, and Komsomolskaya-Magnitnaya. Data for Novinka, Deimos, and Komsomolskaya-Magnitnaya are partly published [12]. Red dashed lines are isolines of $Fe_2O_3$, and the dashed curve separates the values typical of the kimberlitic ilmenites according to [28].

In other Upper Muna pipes, Mg varieties are relatively widespread compared to Zapolayrnaya. The Ti-Mg-Cr-rich varieties are higher in NiO. The more differentiated and low-TiO$_2$ ilmenites split into two branches with different Cr$_2$O$_3$ contents. The Al$_2$O$_3$ content decreases in three linear trends from 54 to 52 wt.% TiO$_2$.

*Chromites* from Zapolyarnaya show a long trend from 68 to 10 wt.% Cr$_2$O$_3$ (Figure 8a). The most Cr-rich varieties (65–61 wt.%), typical for diamond inclusions [29], are mainly low-Ti, but those lower in Cr belong to two trends. The Fe-Ti branch shows enrichment in TiO$_2$ up to 11 wt.%, together with an increase in NiO content with a general decrease in V$_2$O$_5$ from 0.5 to 0 wt.%. The other pipes have similar variations but with far fewer analyses (Figure 8b). In other Upper Muna pipes, Mg varieties are relatively widespread compared to Zapolayrnaya. The Ti-Mg-Cr-rich varieties are higher in NiO.

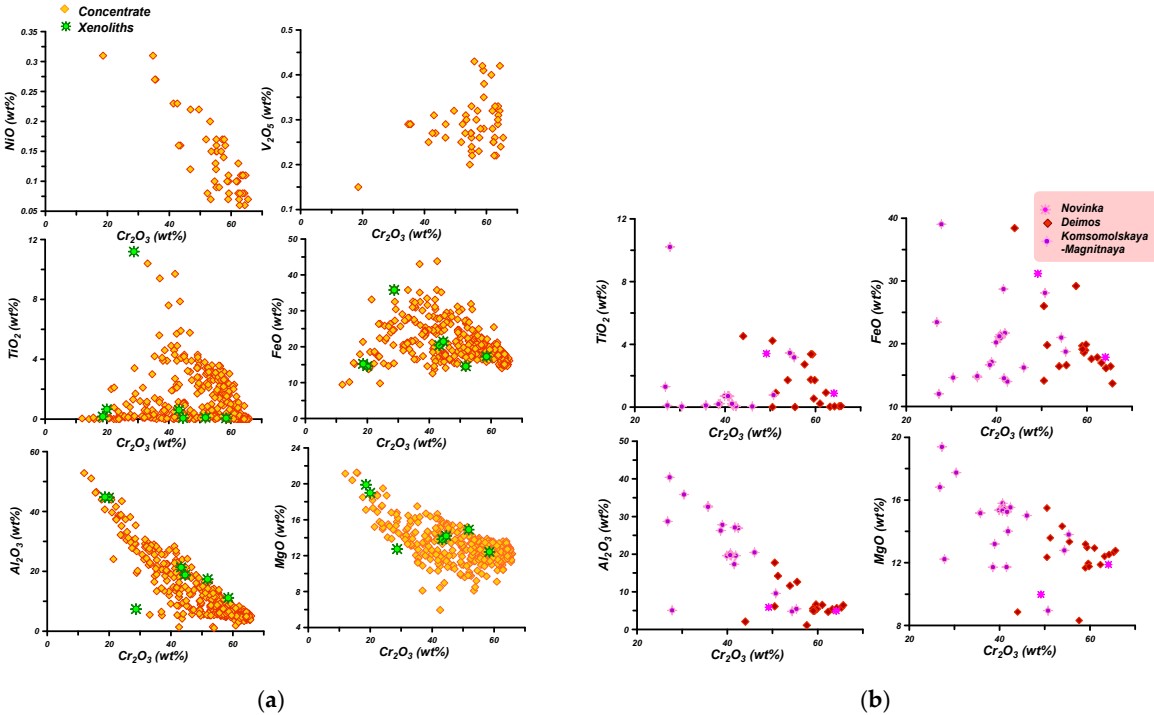

(**a**)　　　　　　　　　　　　　　　　(**b**)

**Figure 8.** Compositions of chromites from Upper Muna field. (**a**) Zapolyarnaya; (**b**) Novinka, Deimos, and Komsomolskaya-Magnitnaya. Data for Novinka, Deimos, and Komsomolskaya-Magnitnaya are partly published [12].

*Amphiboles*, which are more common in the concentrate from the Deimos pipe, are all Cr-Ca hornblendes and pargasites with low Ti contents. They may be subdivided into two groups where K$_2$O, Cr$_2$O$_3$ and Al$_2$O$_3$ slightly decrease together with increasing FeO content (Figure 9).

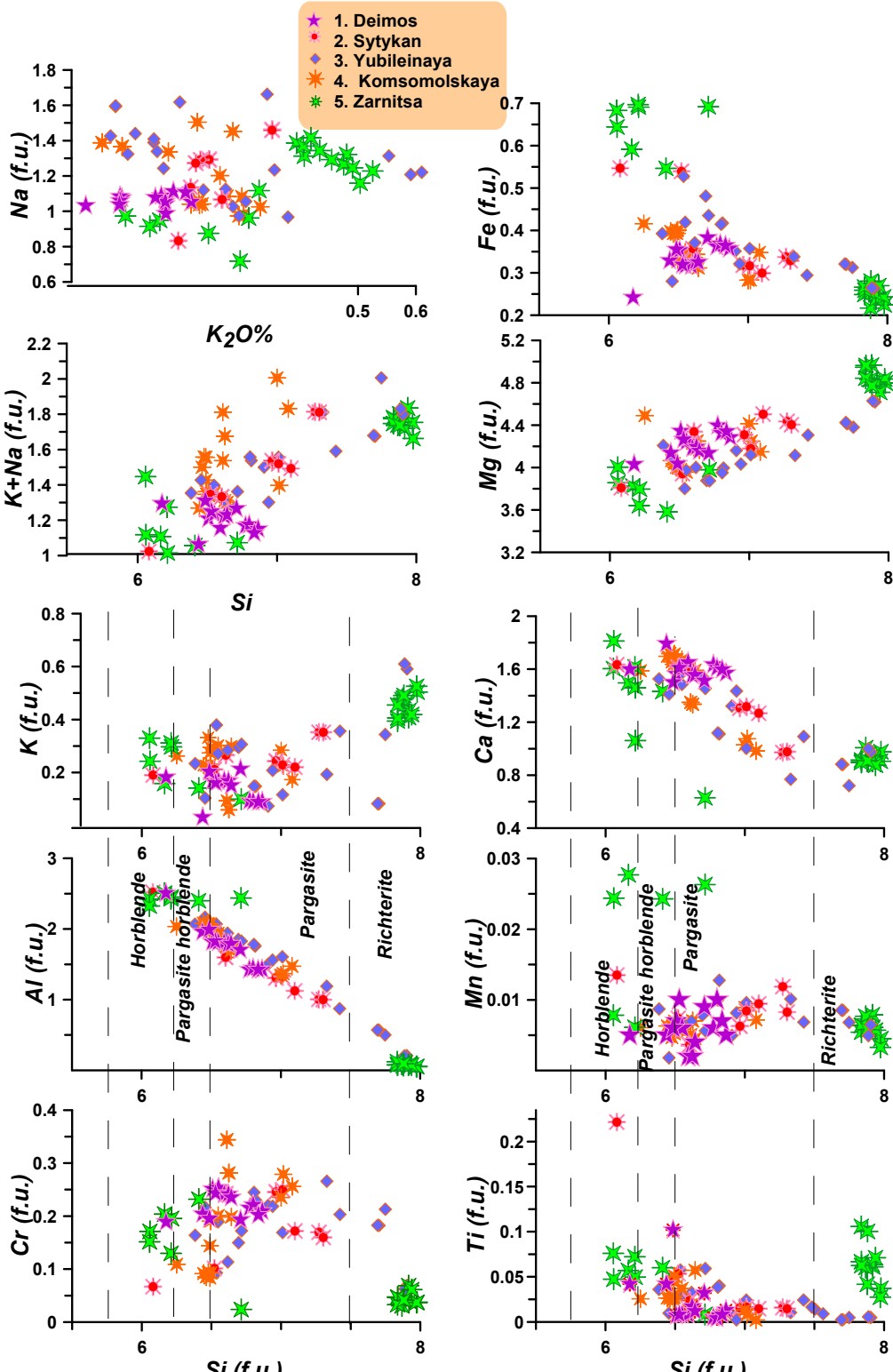

**Figure 9.** Compositions of amphiboles from Daldyn and Alakit fields and Upper Muna field. Data are partly published [2,3,12].

## 6. Geochemistry of Minerals

REE patterns for garnets from the Zapolyarnaya xenoliths show S-type for ~50 % of the analyzed population, which is common for dunitic associations [30,31]. These patterns have minima from Er to Tb and they rarely reveal pyroxenitic slightly concave-up patterns (Figure 10a). They have

HFSE enrichment (mainly Zr, Hf and Nb, Ta), and even slightly elevated LILE. Commonly the HFSE show correlating troughs and peaks. Those with high REE contents have significantly higher Zr, Nb, and Th peaks, suggesting hydrous Phl-bearing metasomatism. Garnets from concentrate also contain similar dunitic garnets and S-type patterns [30,31] (Figure 10a), but they show lower HFSE and LILE abundances. Two pyroxenitic garnets show peaks at Zr-Hf. Common lherzolitic-harzburgitic garnet varieties practically all have more elevated Ta and fluctuating Nb contents.

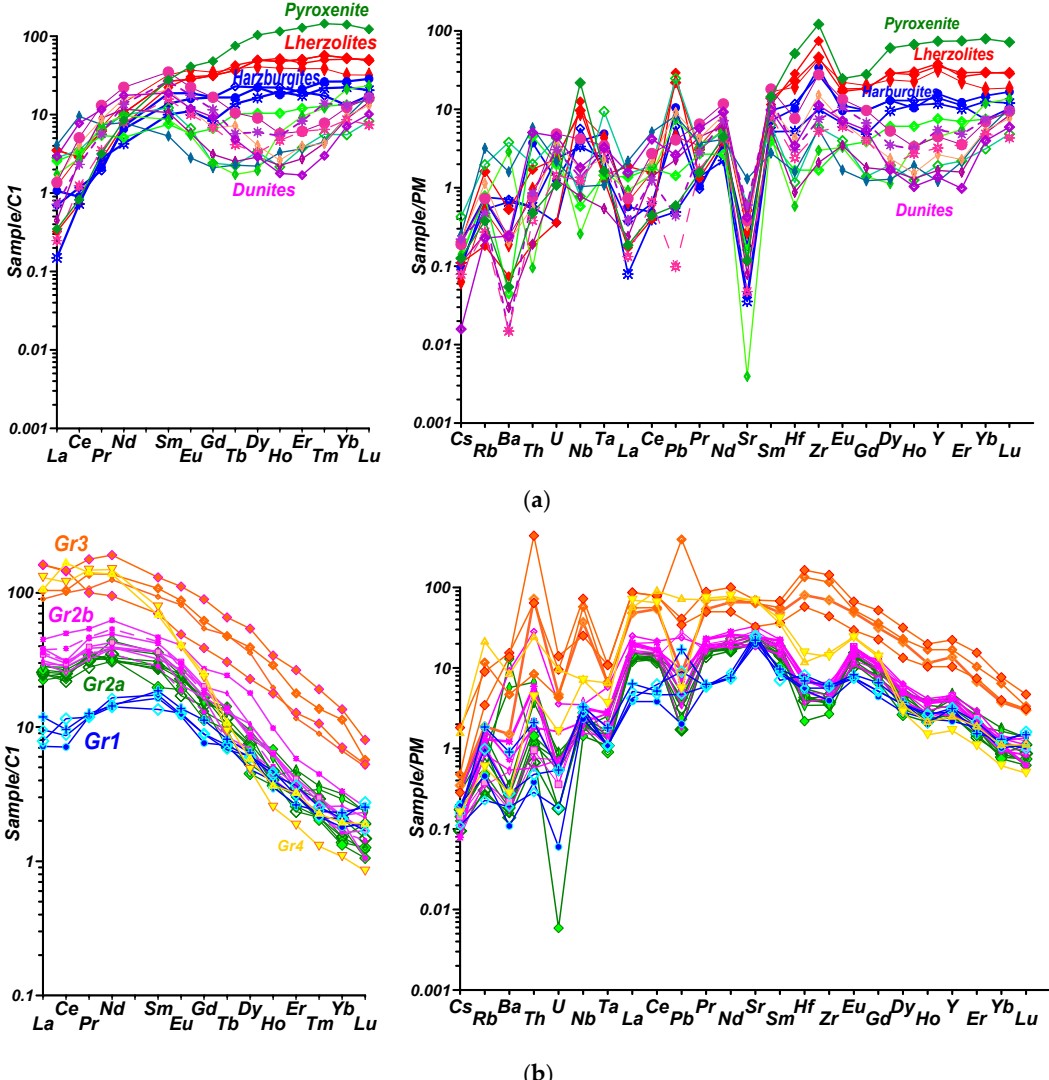

**Figure 10.** REE and spider-diagrams for (**a**) for Cr-bearing garnets, (**b**) Cr-diopsides from mantle xenoliths of Zapolyarnaya pipe. Normalization to chondrite C1 [17] and primitive mantle [32].

The Cpx from xenoliths reveals division into four groups (Figure 10b). Three of them have similar asymmetric bell-like REE patterns like those from megacrysts from the Dalnyaya pipe [26] with increasing REE level (100 to 10/C1) and $(Gd/Yb)_n$. Pyroxenes from the first group with lower trace element contents have semi-round inclined trace element patterns $(Gd/Yb)_n$ ~10 with minima in Ta, Zr, Hf, and Pb and local small peaks of Sr, Nb, and Th. Pyroxenes from group 2 have in general higher trace element patterns without Sr peaks. Pyroxenes from the third group have much higher trace element contents and show high Zr, Hf and peaks in Nb, Th and sometimes in Pb. The fourth group of Cpx has the highest inclination of REE $(La/Yb)_n$ > 20 and a flatter part from Sm to La. They have negative HFSE but elevated Rb, Ba Th, and U, which are near the level ~10 relative to chondrite C1 but lower than La.

Cr-garnets from Novinka pipe (Figure 11a) are quite variable. There are several dunitic garnets with S-type REE patterns. They show U, Th peaks but minima in Zr, Hf. Peridotitic garnets reveal common semi-round REE patterns. Most of them have elevated Zr, Hf contents correlated with the rise of LILE (mainly Rb) and slightly lower peaks in Nb and Ta.

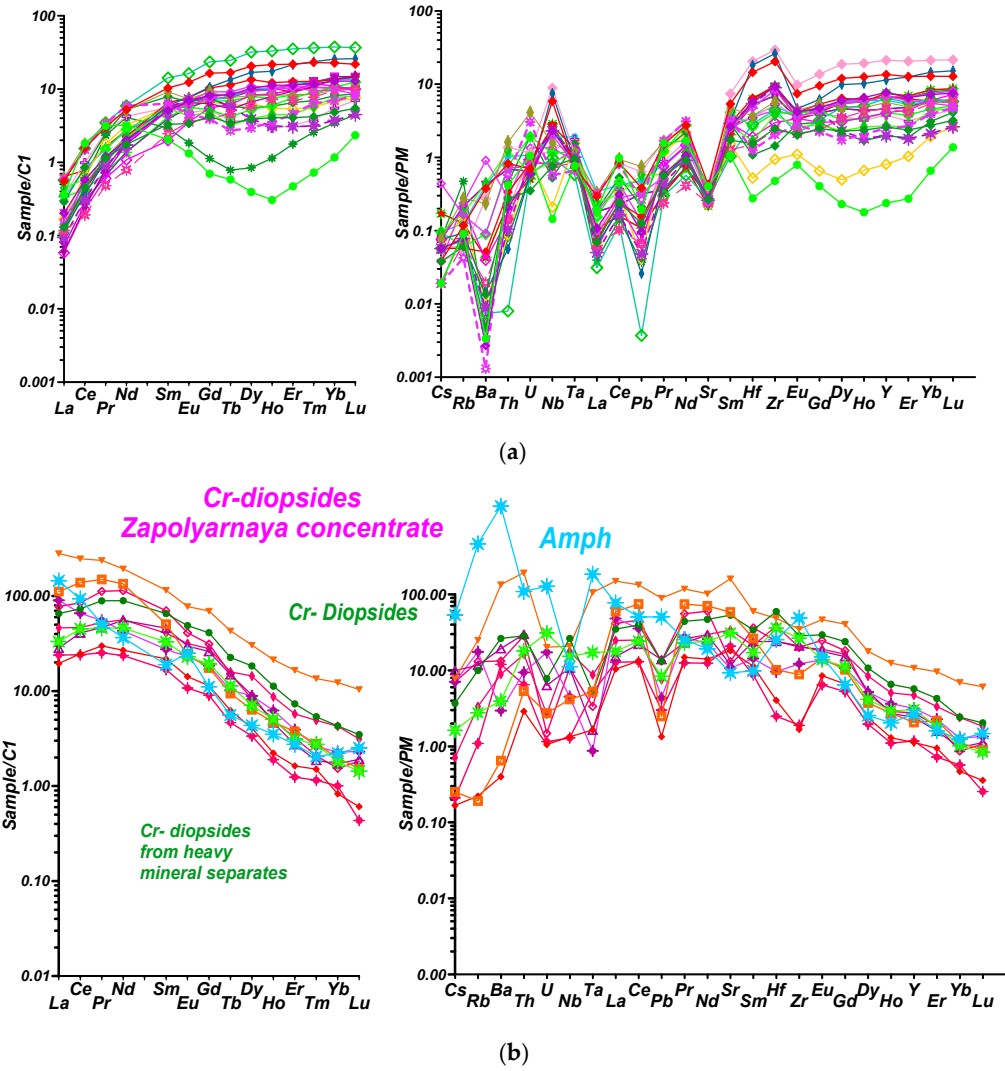

**Figure 11.** REE and spider-diagrams (**a**) for the Cr garnets, (**b**) from concentrates of Zapolyarnaya pipe. Normalization to chondrite C1 [17] and primitive mantle [32].

Cpx from the concentrate of Zapolayrnaya pipe shows slightly different but also inclined REE patterns with less curved slightly concave patterns from La to Nd. The Zr-Hf and Ta-Nb minima are found in two Cpx with lower REE contents. The others have more elevated HFSE at the same level as REE. Ba and Th peaks are found for more enriched varieties (Figure 11b). One amphibole found between Cpx shows a straighter inclined REE pattern and spider-diagram with elevated HFSE and peaks in Rb, Ba, Sr, and U, but the HFSE and Th show minima.

Garnets from Novinka pipe concentrates have fewer dunitic varieties (Figure 11a). They show more asymmetric S-type patterns with peaks in Pr-Nd and peaks of HFSE for the REE-enriched varieties. The U peaks probably reflect the subduction-related features [33]. The common garnets with nearly flat REE part from Sm to Lu or slightly inclined show an increase in the HFSE: Zr > Hf and Nb > Ta.

The Zr peaks suggest an origin for the Cpx, which requires more hydrous metasomatic conditions. In general, this reflects polybaric interaction of the evolving proto-kimberlite melt with the depleted

peridotite mantle. The focus of this interaction was beneath the Zapolyarnaya pipe. Clinopyroxenes from Novinka pipe from group 1 to 3 (G1 to Gr3) demonstrate spreading fan-like patterns in LREE, starting from Sm to La. This spreading in the left side of the spider diagram is even higher with increasing LILE and Th, U values while Zr and Hf are continuously increasing (Figure 11b).

The first most abundant group has moderately inclined REE patterns $(La/Yb)_n$ ~10–15 with a gentle hump at Nd-Pr $(La/Sm)_n$ < 1. Those with lower REE concentrations are also depleted in general in the left part of the spider diagram. The Cpx from Gr2 shows the highest LILE, U, and Th values and slightly elevated LILE. Cpx from groups 1–2 (Figure 12) have small depletions of Zr < Hf as well Ta and Nb. The Cpx from Gr3 have more pronounced Ta minima and the lowest LILE and incompatible element concentrations. They show the most strongly inclined REE patterns with an inflection at Nd and a flat LREE part.

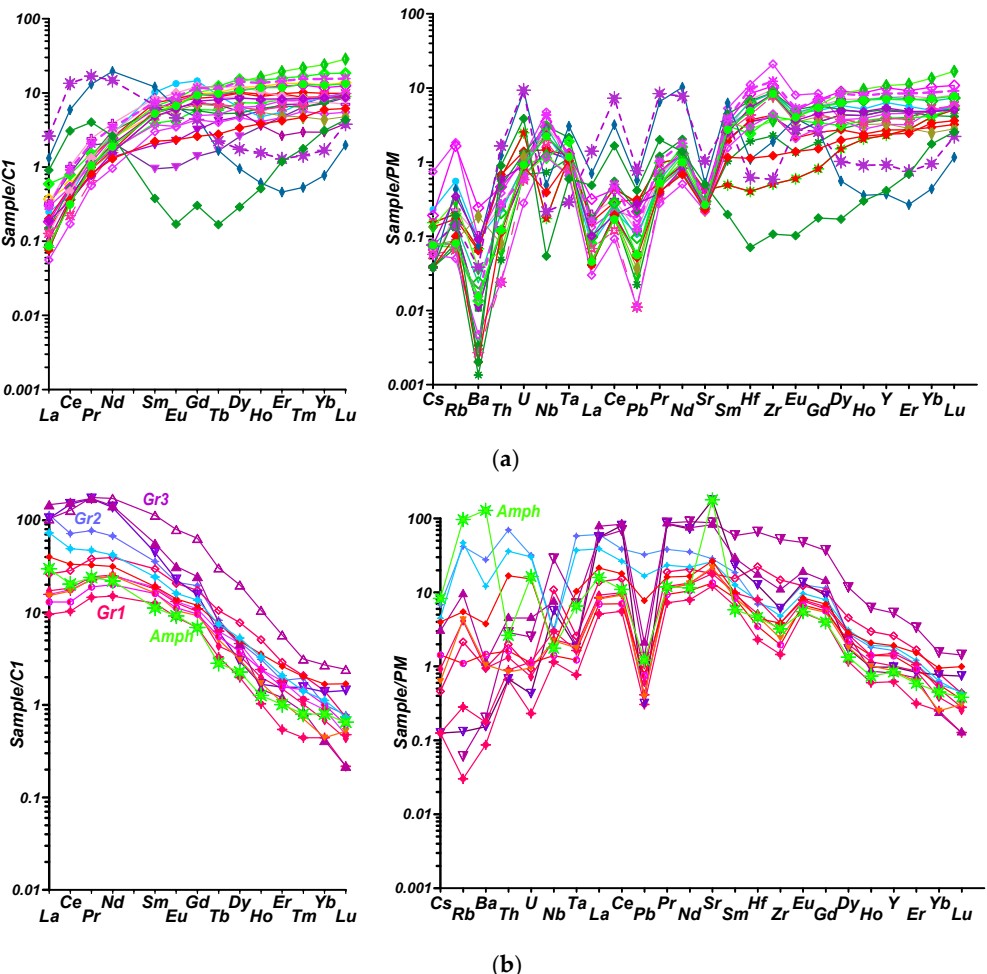

**Figure 12.** REE and spider-diagrams (**a**) for the Cr garnets, (**b**) for the Cpx from concentrates of Novinka pipe. Normalization to chondrite C1 [17] and primitive mantle [32].

## 7. PT Reconstructions and Mantle Layering

PT estimates and geochemistry minerals for 50 xenoliths from Zapolyarnaya pipe were obtained for the first time. All the PT reconstructions were made using monomineral versions of thermometers [12,26,27,34–41] for garnet, clinopyroxene, orthopyroxene, chromite, and ilmenite compositions. Details of the calibration of the thermobarometers can be found in previous papers [12,37,41]. Commonly the precisions of the methods are ~2–5 kbars, but relative precision is better. For the clinopyroxene it is better and is near 2 kbar, so the method is comparable to [36].

However, it works for peridotitic, basaltic and eclogitic associations. The results of the methods mutually coincide.

The garnet geotherm (Figure 13) is located at 38 mW·m$^{-2}$ at LAB (6 GPa) and crosses the conductive geotherms to reach the Moho at 600 °C. Below the LAB it extends to 8 GPa splitting into low temperature (35 mW·m$^{-2}$) and high temperature (45 mW·m$^{-2}$) branches.

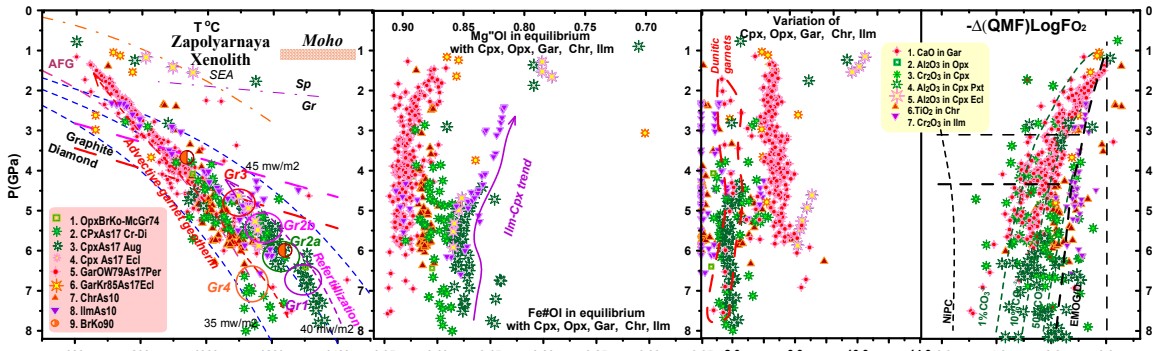

**Figure 13.** P-T-X-fO$_2$ diagram for the minerals from deep-seated xenoliths from Zapolyarnaya pipe. Some xenocrysts from essentially "dunite veins" have been included. Symbols: 1. Opx: T °C-[34]-P(GPa) [35]. 2. Cpx: T °C-[36]-P(GPa)-[37] (for Cr-diopsides); 3. The same for Fe-rich Cr-diopsides. 4. The same for eclogites and pyroxenite). Garnets: 5. T °C-[38]-P(GPa)-[37] 6. The same for eclogites; Chromite: 7. T °C-[39]-P(GPa)-[12]; 8. Ilmenite megacrysts [40]-P(GPa)-[41]; 9. T °C-P(GPa)-[34]. The field for P-fO$_2$ (ΔLogfO$_2$ relative to quartz-fayalite-magnetite QMF buffer) diagrams after [42,43]. The oxybarometers for Sp and Ilm [40]; Cpx [44]; Gar [45]. The horizontal dashed line at 3.5 and 4.5 GPa corresponds to the Graphite-Diamond boundary [46] at 35 and 40 mW·m$^{-2}$ respectively. The upper line the same boundary after [47]. Conductive geotherms after [48]. The South Eastern Australian geotherm (SEA) after [49]. The garnet spinel transition is after [50]. Moho boundary (brown rectangle) after [51]. Abbreviations in legend: Per-peridotites; Ecl-eclogites; Aug- augites; Cr-Di- Cr-diopsides. Methods abbreviations—see legend and literature references. The values of Mg'Ol and Fe'Ol mean the Mg' and Fe' numbers for the Ol in equilibrium with the minerals are calculated according to [12]. For the marked geochemical groups of Cpx and Gar, see Figure 10b.

In the upper part it becomes sub-adiabatic, crossing conductive geotherms. The garnet trend became more Fe-rich in the upper (<4 GPa) sub-cratonic lithospheric mantle (SCLM). The alternative garnet thermobarometer [52,53] always locates all the PT points near the diamond-graphite boundary except for sub-Ca garnets. Commonly they project Ni temperatures onto the 40 mW·m$^{-2}$ geotherm [54].

Most Cpx from the xenoliths are of refertilized types like those from deformed or porphyroclastic peridotites from Udachnaya [20,55,56], with newly formed grains of pyroxenes and sometimes garnets in an essentially olivine matrix. Together with the ilmenites, they also yield a high-T geotherm, tracing a convective branch (40 mW·m$^{-2}$) with an inflection to a hot branch at 6 GPa and continued to 8 GPa. A low-T branch for the Cr-rich varieties also exists. The Cr-rich low-Fe pyroxenes mostly plot on the 36–38 mW·m$^{-2}$ geotherm together with the Cr-spinel estimates from 6 to 3 GPa and reflect the low temperature conditions in the lower part of the mantle section. The ilmenite tend is very long, extending from 7 to 2.5 GPa, and is accompanied by increasing Fe# from 0.11 to 0.17.

The Cpx split in the P-Fe# trend. The middle values are common to the ilmenite trend, whereas the lower one is intermediate between the Fe# of garnets and ilmenites and the higher one is a pyroxenitic trend overlapping the Ilm trend from 6 to 4 GPa and with the trend for Al-Na-rich pyroxenes. The highly Fe-rich pyroxenitic Cpx corresponds to the Moho boundary, which is placed at 1–1.2 GPa [51]. In the deeper part of the SCLM, Cpx are essentially more oxidized and closer to fO$_2$ values of ilmenites. In the upper mantle section, they relate to more reduced conditions close to the garnet trend. Common Cr-diopsides are related to the oxidation of garnet trend [57] (10–20% CO$_3^{-2}$ in the coexisting

melt) [43]. The pyroxenitic Fe-enriched pyroxenes also mainly plot in the lower SCLM part in the relatively reduced conditions favorable for the diamond stability.

The variations of the P-CaO garnets show the dunitic garnets trace practically the entire SCLM from 1.8 to 7.8 GPa, and there are several fluctuations of CaO with depth, reflecting mantle layering. The pyroxenitic varieties appear deeper at 6.0 GPa.

The P-T-X-fO$_2$ diagram for Zapolyarnaya based on KIM from concentrate (Figure 14a) shows nearly the same geotherm for garnets. Mantle section is layered according to garnets showing several Ca-rich fluctuations in the P-CaO trend. At the same time, a number of low-Ca values for sub-Ca garnets are found from 2 to 8 GPa. Cpx that show a common peridotitic signature range from the lithosphere base (LAB) to 1.5 GPa. Ilmenites of Fe-Ti type give a rather complex trend from the LAB to 2.5 GPa. However, there is also a separate Mg-rich trend corresponding to ilmenites formed in dunitic veins (Fe#Ol = 0.07), also tracing the entire lithosphere thickness. The Cr-rich (to 5 wt. %) ilmenites found from 6 to 3 GPa suggest intense protokimberlite metasomatism in the mantle column, because the ilmenites derived from protokimberlites are commonly Cr-free [41]. Chromite PT values give a rather wide geotherm close to those determined for garnets. Ti-rich varieties are more frequent in the lower part of the SCLM. The pyroxenes and ilmenites from Zapolyarnaya pipe mostly plot in the diamond stability field in the P-fO$_2$ diagram. Only the deepest varieties are more oxidized.

The nearby Deimos pipe reveals rather different thermal conditions having a lower (35 mW·m$^{-2}$) geotherm at 6 GPa and at about 8 GPa (Figure 14b). There are increasing FeO levels at 3.5, 4.8 and 5.5 GPa. The amount of dunitic garnets is higher in the middle and lower parts of the SCLM but the pyroxenitic trend is also pronounced in the lower SCLM from 4 GPa. There are at least three types of Cr-diopsides: The first is in the middle part, close in Fe#Ol to Cr-garnets from peridotites (lherzolites), the 2nd trend is intermediate between ilmenitic and peridotitic trends, and the third is close to the ilmenite fractionation trend. Typical eclogitic Cpx form the small trend from 5 to 3.5 GPa. The ilmenite trend also has three branches: The first is low-Cr at 7–6.5 GPa; the second has moderate Cr from 5 to 3.5 GPa; and the third is Cr-rich to 2 GPa related to the Amph-Phl metasomatites.

In the PT diagram for Novinka pipe, the garnet geotherm is similar to those from the previous pipes but at slightly lower temperatures (Figure 14c). It is divided into several (four large) sharp intervals with positive inclinations of the P-Fe# trends. Garnets also show a depletion trend starting from 2 to 7.5 GPa and, at the LAB at 7 GPa, the garnet trend in the P-CaO plot splits into pyroxenitic and dunitic branches (Figure 14c). This mantle column contains more fertile material than that beneath Zapolyarnaya and Deimos. Cr-diopsides vary in Mg# from 0.93 to 0.84, showing three definite trends in P-Fe#. The first is for common lherzolites. The second is close to the ilmenite trend. In addition, there is a more Fe-rich branch that probably appeared after reactions with Fe-rich eclogites.

The P-T-X-fO$_2$ diagram for the Komsomolskaya-Magnitnaya (Figure 14d) pipe is similar to those determined for Zapolyarnaya and Novinka (Figure 14c). We added Cpx compositions from [15] and found the same inflection of the Cpx geotherm at 6 GPa, though the other thermobarometric methods yield even higher temperature conditions to 1450 °C [15]. Most pyroxene PT estimates are in the 3.0–6.5 GPa interval and correspond to the 40 mW·m$^{-2}$ geotherm, which is supported by the Gar-Opx based method [34]. The pyroxenite branch exists only in the deeper part of the mantle column at 7–5 GPa and corresponds to the Ilm trend in Fe# values (Figure 14d). The garnet geotherm in the lower part of the mantle section splits into the high temperature branch 44 mW·m$^{-2}$ for the lherzolitic and pyroxenitic varieties and low temperature branch close to 37 mW·m$^{-2}$. The Opx-based [35,36] methods also give similar conditions for the hot geotherm branch.

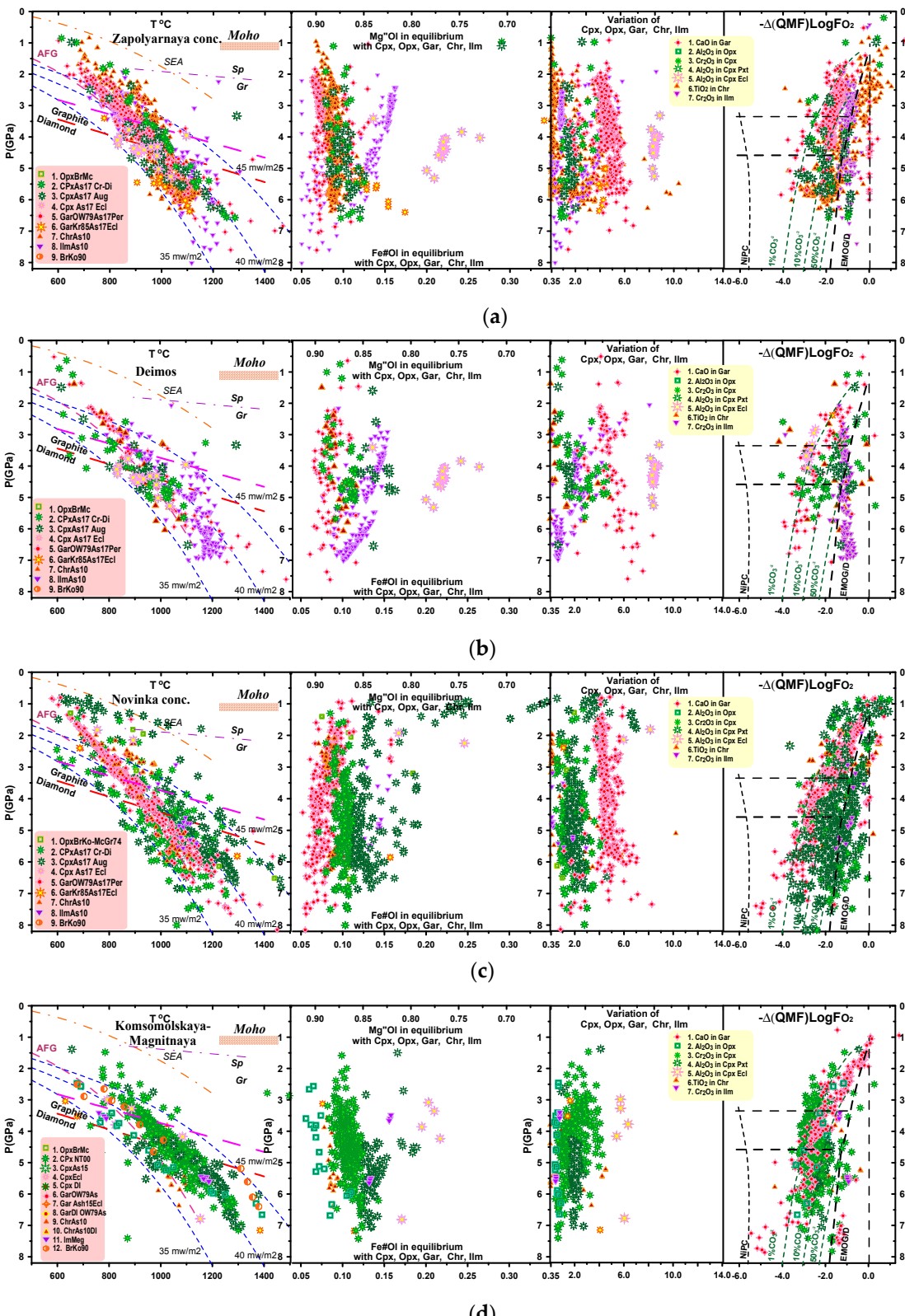

**Figure 14.** P-T-X-fO$_2$ diagram for the concentrate from (**a**) Zapolyarnaya pipe. (**b**) Deimos pipe. (**c**) Novinka pipe. (**d**) Komsomolskaya-Magnitnaya pipe. The symbols are the same as for Figure 10.

## 8. Discussion

### 8.1. Mantle Layering

The regular fluctuations of CaO and FeO of garnets and rhythmic compositional features of the other minerals with the depth allows us to suggest that the mantle layering was caused by the accretion of subducted plates to the base of the craton keel [58]. This assumption is supported by the seismic tomography models. However, there is also a model of stacked slabs with cratonic growth from the margins [59]. We suggest the compositional fluctuations with depth as repeated mantle layers. However, everywhere in the mantle columns we see the spreading pyroxenitic and dunite harzburgitic arrays to the base of the SCLM. This suggests melting of peridotites [60] and more probably eclogites and differentiation of the mantle melts with pervasive melt percolation, which started at depths of 6–5 GPa after subduction and accretion of the slabs according to the bulk rock estimates [55,60,61]. Several such melting models for eclogites and peridotites were published [60–62].

In the lower part of the SCLM, the garnet trend spreads towards the pyroxenitic and dunitic branches. This is a common feature of the mantle beneath the Siberian craton [3,12] and worldwide. It may suggest differentiation in the lower SCLM due to modal melting in early stages [62] or later under the influence of carbonate-$H_2O$ fluxes from subducted slabs [63]

We suggest a model of low-angle subduction in the presence of a superplume [58]. The ultramafic plume could not rise from depths of 8.0 GPa because it was too dense [64] and needed additional differentiation, carbonation or hydration to cross the dense SCLM. It took place due to the interaction of the hydrated and carbonated subduction fluxes with the base of the SCLM. The subducted slabs experienced melting and were mostly eclogites, which were hybridized with peridotites and/or moved to the surface creating specific pyroxenites and remelted eclogites and forming the lens in the middle of the mantle section [65]. The number of layers in the deep mantle lithosphere is close to seven or six everywhere in the world, which is close to the number of superplume events from 4.2 to 2.7 B.y. [66]. The energy of the common plumes is insufficient to melt not only eclogites, but peridotites also. Eclogite melting stops the subduction. As the source of Phl and amphibole may not be only distant subduction-related fluids and subducted sediments, reactivation of metasomatically modified subduction wedges may have been involved in peridotite melting at the continental margins, resulting in strongly depleted peridotites in the continental margins.

The rapid growth of the continental lithosphere at 2.7–2.8 [66] was accompanied by generation of abundant granites and general oxidation of the mantle and the appearance of a great amount of $H_2O$ in the mantle, accompanied by the creation of dunitic channels where the diamonds may have grown [67,68]. This occurred 2.7 B.y. ago before the great oxidation event in the crust. It was accompanied by the appearance of alkaline melts and decrease of mantle viscosity, together with acceleration of convection and the onset of rapid subduction dynamics. However, many diamonds formed during the re-melting of eclogites and associated carbon-rich subducted sediments [69]. However, there are data that indicate that megacrystalline dunites [22] are more ancient than harzburgites [70].

The layering is clearer in the SCLM beneath Deimos or Komsomolskaya-Magnitnaya pipe where melt percolation was not as intense as beneath Zapolyarnaya and Novinka. Such pervasive melt percolation was also found beneath Zagadochnaya [71] and Novinka [14]. Such waves of melt migration should have caused a compositional transformation on the boundary between the subduction slabs. However, there is also a model that melt percolation with reactions on phase transition physico-chemical boundaries could cause the pseudo-layering [72]. The middle SCLM corresponds to the inflection and minima of the peridotite solidus in the presence of volatiles. Thus, segregations such as the pyroxenite lens in the middle part of craton keel are possible [65].

According to the PT estimates, we could also suggest the presence of an eclogite lens in the middle of the SCLM beneath the Deimos pipe.

In many cases, there is evidence of changes in the P-Fe# trends for Cpx and garnets and their inclination after superplumes [58], for example after the Siberian superplume, which suggests pervasive melt percolation at least at the weakened zones.

The P-T-X layering that was reconstructed with mineral thermobarometry shows the presence of at least two or three stages of melt percolation from the Cpx P-Fe tr-*ends (Figures 13 and 14) for the Deimos pipe (Figure 14b). The trends of the chromites with strong enrichment in the ulvospinel component may also suggest interaction with Ti-rich melts that produce ilmenite megacrysts [38,73]. Several pipes in Upper Muna and Prianabarie [74] show the preferential distribution of ilmenite or chromite in the concentrate, as well as in Daldyn and other regions.

In some cases, we suggest crystallization of chromites instead of ilmenites, which is more common for micaceous kimberlites richer in $H_2O$. Zapolyarnaya and Poiskovaya pipes are richer in Phl and $H_2O$ and chromites.

## 8.2. Depletion and Regeneration Via Modal Metasomatism

In general, the xenoliths and mantle columns beneath Upper Muna and especially the Zapolyarnaya pipe, demonstrate a general depletion of the whole mantle column, whereas in the Daldyn-Alakit [2,3] and Malo-Botobinsky regions [6,11,26] as well as in Arkhangelsk region [75,76] this depletion is typically confined to the lower SCLM. Presence of Cpx veins and veinlets in Zapolyarnaya xenoliths suggests refertilization, which is described from kimberlitic mantle xenoliths worldwide [17,58]. However, they are rather Fe-rich, suggesting plume-related metasomatism [77]. Moreover, such Cpx are found in association with Ilm megacrysts [72] in South Africa and the Dalnyaya pipe [26]. The Cpx path in PT diagram for Zapolyarnaya pipe exactly follows the ilmenite trend corresponding to the protokimberlite-derived association. Together with the ilmenites, they show the lowest trace element values and probably are derived from melts concentrated in the dunite veins, which served as melt conduits [78]. These channels were probably created in the subduction stage during the hydration of mantle and acceleration of convection in mantle ~3.0–2.7 B.y. [66] and the last event of craton growth when a great amount of $H_2O$ fluids passed through the cratonic mantle lithosphere causing garnetization and creation of the garnet semi-advective geotherms [79] and modal mantle metasomatism [80], which may be accompanied by increases in trace elements [81].

Modal mantle metasomatism with amphibolization of the upper part and appearance of Phl scattered and veined metasomatism throughout the SCLM is also typical of the Upper Muna region. This is somewhat similar to the Alakit region where amphiboles and phlogopites are common in mantle xenoliths, showing wide variations in compositions [3,82]. They are also common in the northern regions of the YKP in Obnazhennaya [83] and Prianabarie [74].

A very wide range of compositions is determined in Leningrad [84] and Sytykanskaya pipes [81]. Among kimberlites worldwide [85], they are found in Kimberly pipe [86], though often they are found in the MARID xenoliths [87].

Phlogopite metasomatism is a more common feature in the SCLM [88,89]. Phl and silica-enrichment are typical for subduction-related melts rich in $H_2O$ [90], which probably caused the reactions with the growth of Phl and amphiboles, which are found in many orogenic massifs [91].

## 8.3. Metasomatism and Geochemistry

The most common garnets show variations of the $(La/Yb)_n$, which is regulated by the temperatures and subsolidus reactions [92]. Higher temperature (and pressures) correlate with lower inclination of REE patterns and low $(La/Yb)_n$. The latter ratio is negatively correlated with $Cr_2O_3$. Similar relations are observed for Cpx. This is the common geochemistry of REE in peridotites.

From the geochemistry, it is clear that there are two types of metasomatism. The early subduction melts/fluids had primary continental signatures. The Zr peaks and high LREE levels for most minerals in the concentrate from the same pipe may mean that these are the relic features of mantle. Commonly the $H_2O$ metasomatism is accompanied by an increase in Zr in garnets [93] and other minerals and also

BaO, SrO, Pb, and LILE [33]. This should be related to subduction-related melts and metasomatites formed by reactivation of early metasomatic associations. Wide distribution of dunites suggests that the mantle beneath the Upper Muna field was permeable also for the subduction-related fluids and melts, creating metasomatic associations enriched in Zr and LILE [93]. That was the ancient metasomatism [90,92].

The increase of Th and Nb may be explained by the regeneration of the mantle lithosphere due to the mixing with the subduction-related material containing rutiles [94] and possibly some rare minerals like apatites. The accompanying sediments may even have included monazite or zircons. The common subduction-related melts show low HFSE levels due to retention in rutile. However, subduction-related carbonate melts could carry a lot of HFSE [95]. The different behavior of HFSE is visible in both garnet and Cpx spider-diagrams. Rutile is effective at decoupling Nb and Ta [96], concentrating Nb, and possibly melting of such eclogites during ancient subduction determined the geochemistry of the Upper Muna mantle minerals. This is clearer for garnets and metasomatic minerals like amphibole and phlogopites, but is less clear for the pyroxenes that were formed under the protokimberlite influence.

One more specific observation is the differences of the dunitic garnets from Novinka, which are very LREE-enriched and have U peaks. Garnets from concentrate and xenoliths from Zapolyarnaya pipe also have U and Nb peaks, probably a sign of ancient subduction-related fluids.

Relative abundance of the lower part of the SCLM in the pyroxene material and their enrichment in Fe and HFSE may suggest the reaction of the lower horizons with the protokimberlite melts, because they have crystallized large amounts of ilmenite [41,73]. The ilmenite trend extends to the upper horizons of the mantle column, and protokimberlite crystallization was accompanied by the deep differentiation of protokimberlites tending to carbonatites [41] and derivation of the metasomatic fluids and of the veinlets in wall-rock peridotites.

The protokimberlite-related metasomatism [77] also may have taken place in several stages as is visible in the complex ilmenite pyroxenite P-T-X paths. Ti-rich garnets are related to the rather hot stages that may correspond to the rising feeding systems for the protokimberlite melts producing an ilmenite vein network, large channels and possibly magmatic chambers. The ilmenite PT estimates trace separate discrete arrays in the mantle columns. From the geochemistry of Cpx and garnets, it is possible to suggest three stages and levels of the protokimberlite melts evolution with the general differentiation and enrichment.

The first is connected with the protokimberlite melts [41,77], which produced the stepped increase of REE and trace elements in Cpx (Figure 11b) and followed the protokimberlite differentiation (Gr1-2) and also partial melting of peridotites (Gr3). In the first two groups, the influence of partial melts is not so clear. The low HFSE level is possibly due to ilmenite crystallization but decoupling in Ta and Nb possibly means the participation of rutile [97]. For Gr2 Cpx, the incompatible elements slightly increase. However, for groups 3 and 4, the influence of material and metasomatic components becomes clearer. The reconstructed melts with the KD (partition coefficients) with the addition of KDs from [98–101] (Figure 15a) show the jagged spider-diagram, which may be derived from essentially oxidized carbonatitic melts, which partly lost HFSE. However, the elevated amounts of components such as LILE, Sr, Th, Rb, Nb, Y, and U may also suggest contamination by some subducted material, which could also incorporate continental sediments [102] and has a strong Zr peak. Further enrichment caused the appearance of the strong peaks and also elevation in Rb and all LILE.

The parental melts for the garnet from the xenoliths (KD after Green et al. [101]) (Figure 15a) are significantly different. Even common garnets have different inclination of REE patterns and mostly slightly concave, and the dunitic garnets reveal the higher inclination with the hump at Ce-Pr and depressions from Gd to Yb. The observed peaks in Rb, Ba, Th, Nb, Pb, and Zr are closer to subduction-type melts. Thus, the garnets and clinopyroxenes in xenoliths are in total disequilibrium. The Cpx were produced later mainly in the refertilization stage after protokimberlite influence.

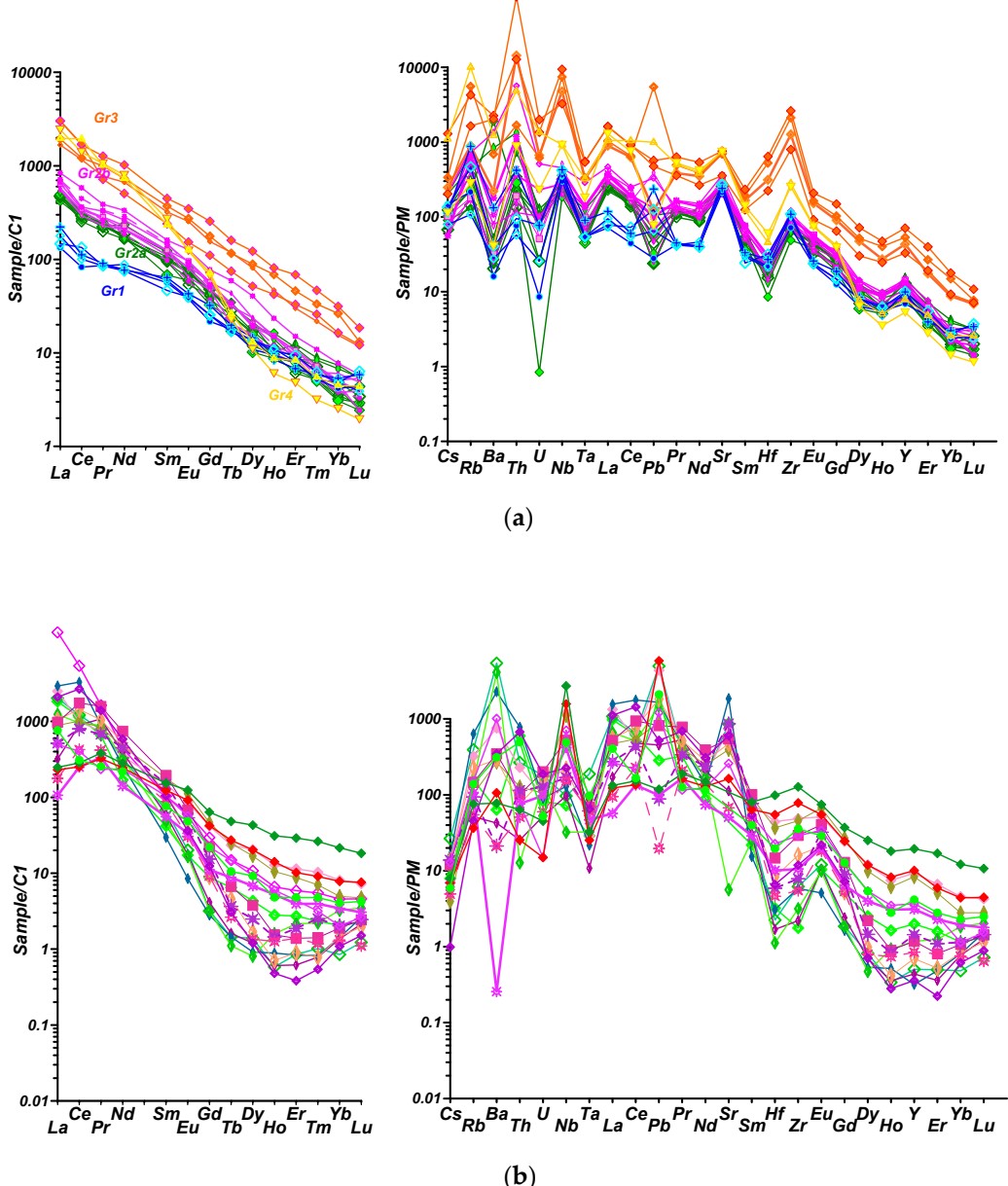

**Figure 15.** REE and spider-diagrams for the hypothetical melts in equilibrium (**a**) with the Cr- diopsides calculated with KD [98] with the addition of [99,100]; (**b**) with Cr-garnets calculated with KD [101] from mantle xenoliths of Zapolyarnaya pipe. Normalization to chondrite C1 [17] and primitive mantle [32].

The key question about metasomatism is the behavior of the HFSE and mostly Zr. As is clear from all the magmatic processes, Zr is an indicator of high activity of $H_2O$ in melts and also of subducted sediments like under the North China craton [102]. Enrichment of the later Cpx in Zr and LILE could be the result of two alternative processes. The first is the differentiation or mixed carbonate-$H_2O$-bearing alkaline melts, which are found sometimes as melt inclusions [103]. The other possibility is the admixture of partial melts in peridotites after melting of Phl-bearing metasomatites, which increased the LILE and LREE. The ultramafic carbonated melts are also a source of HFSE enrichment [104].

The patterns of Cpx from the Novinka pipe are more controversial. The most REE-rich compositions of Gr3 have very low incompatible elements and slightly lower in REE. The Cpx from Group 2 have the highest incompatible element abundances including REE. It means that the Gr2 crystallized from the melt mixed between the peridotite metasomatites and protokimberlites. In addition, Cpx from Gr3 includes pure derivates from protokimberlites and contaminated varieties.

Comparison of the REE and spider-diagrams for minerals from the YKP [3,14,55,71,74,75,77,80] shows that similar processes are common in mantle columns beneath the Siberian craton.

### 8.4. Relations of the Dunites, Diamonds and Protokimberlite Metasomatism

The abundance of depleted peridotite and sub-calcic garnet (G10) associations is rather high for the perspective of diamond production [67]. Moreover, the dunites served as melt conduits [78], although the protokimberlitic melt chose Zapolyarnaya pipe as the main melt conduit.

The mineralogical features as well the structure of the mantle columns of the Upper Muna mantle slightly differ from those of the other regions of Siberian platform. Widespread enriched Cr-diopsides that are found in diamond inclusions are not very common for the Daldyn and even Alakit region. The rather low abundance of ilmenites is more similar to the Nakyn field than for the central region. The structure of the mantle columns shows also the abundance of pyroxenite material in the lower mantle lithosphere. Geochemistry of the minerals shows the high distribution of the volatile-related components like LILE, including Ba typical for subduction processes [33]. The influence of fluids during crystallization of the diamonds is suggested by the decrease in average $\delta^{13}C$ (−4.59 ‰) and by relatively low average N contents. Temperatures for Zapolyarnaya diamonds are around 1100–1200 °C [8]. This is mainly related to the ancient rounded partly dissolved diamonds.

Heating of the lower SCLM is more pronounced beneath the Zapolyarnaya pipe and it is suggested to be more prospective for large good quality diamonds because many large pipes like Premier [105], Orapa [106], Zarnitsa [3], Mir, and Udachnaya [55,93] show very heated LAB and deep roots. Thus, the large diamonds that are found in this pipe (Figure 3c) may be type II megacrystic diamonds [107]. In Zapolyarnaya, the rare largest diamonds show beautiful shaping and a lack of dissolution signs, and probably were created at the latest stages. Dating of diamond inclusions from Siberia commonly gives Archean and Proterozoic ages [108]. However, there is also isotopic evidence for diamond creation close to age of eruption [109]. The appearance of the deformed peridotites in the LAB beneath the Upper Muna field [15] in turn was formed by reaction with the oxidized protokimberlite melts [110] and has a minimum in mechanical strength at 6.0 GPa [111]. After their interaction with the peridotites, the melts became more reduced as seen in the clinopyroxene geochemistry. In such weakened and rather reduced conditions buffered by peridotites, the growth of diamonds is possible.

The opinion that some kimberlites are relatively low temperature magma [112] does not apply to the Upper Muna, because there the kimberlites are MgO-rich, which could just reflect mantle xenocryst/xenolith load, rather than MgO content of kimberlite magma near the LAB. Protokimberlites produce rather high temperature interaction at the LAB. However, the conclusion that the geochemistry of the kimberlites is determined by the ambient mantle lithosphere [113] is realistic. In this case, the subduction-related components may be found in protokimberlites.

The large well-shaped diamond should be related to the protokimberlite event. The other types of prevailing diamonds with slightly rounded forms may be ascribed to resorption due to instability with a carbonated melt. According to experimental data, the presence of $H_2O$ causes rapid diamond crystallization [114] and complex shapes of the crystallized diamonds [115]. Moreover, it was demonstrated that, for diamond crystallization at 7.5–6.3 GPa at the LAB, the temperatures should be 1450–1570 °C [116], which just corresponds to the conditions of the protokimberlites at the LAB, and conditions determined for the SCLM beneath the Zapolyarnaya pipe. Presence of Phl in diamonds [117] supports this idea. Alhough the diamonds of course could have different times of origin in different conditions [32], now more information is being published about the formation of diamonds close in time to protokimberlite processes [117].

The melts in equilibrium with Cpx are similar to protokimberlites in REE elements, but they essentially mixed with metasomatic material. This was accompanied by increasing oxidation of Cr in Cr-diopsides and increase of all incompatible elements and general reduction in $fO_2$ during the rising of protokimberlite melt and reaction with the mantle.

### 9. Conclusions

1.  The Upper Muna field includes four diamondiferous pipes: Zapolyarnaya, Deimos, Komsomolskaya-Magnitnaya, and Novinka, which contain deep materials typical of the continental mantle keel, very rich in the depleted varieties and subjected to subduction-related phlogopite metasomatism in ancient times.

2.  The geochemical features with Zr and Th peaks and enrichment in LILE of peridotite minerals show rather enriched continental or metasomatic signatures.

3.  Multistage processes of melt percolation recorded at the PT paths in mantle columns suggest a lithospheric mantle structure that was permeable for melts.

4.  The largest diamonds as well as other good quality diamonds may result from the crystallization of protokimberlite melts highly contaminated in peridotites or mixed with partial melts from metasomatized peridotites.

5.  The structure and mineralogical features of Zapolyarnaya pipe and their trace element signatures are prospective for diamond exploration.

**Supplementary Materials:** The following are available online at http://www.mdpi.com/2075-163X/10/9/755/s1, Table S1: EPMA for minerals from xenoliths and concentrates, Table S2: TRE in minerals from xenoliths and concentrates.

**Author Contributions:** Sample acquisition, I.A. and N.V.; Methodology, N.M., I.A. and A.I.; Validation, I.A.; Formal analysis, N.M.; Investigation, I.A. and N.V.; Resources, I.A.; Data curation, N.M., I.A.; Writing—original draft preparation, I.A.; Writing—review and editing, I.A. and H.D.; Visualization, I.A.; Supervision, I.A.; Project administration, I.A.; Funding acquisition, I.A. All authors have read and agreed to the published version of the manuscript.

**Funding:** Ministry of Science and Higher Education of the Russian Federation. Supported by RFBR grants 19-05-00788a; 18-05-00073a; 16-05-00860a. Work is done on state assignment of IGM SB RAS and the governmental assignment in terms of Project IX. 129.1.4.

**Acknowledgments:** To Gleb Shmarov and geological stuff of ALROSA company. To the four reviewers for the reviewing and their helpful comments.

**Conflicts of Interest:** Authors declare no conflicts of interests with the other researchers and organizations. Data are mainly original. We used also data for diagrams from the published papers.

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
