# Peer review of "Thermobarometry and Geochemistry of Mantle Xenoliths from Zapolyarnaya Pipe, Upper Muna Field, Yakutia: Implications for Mantle Layering, Interaction with Plume Melts and Diamond Grade"

_minerals, doi:10.3390/min10090755_

Round 1
Reviewer 1 Report
This paper reports major and trace element concentrations for mantle xenoliths and concentrate-derived minerals from various kimberlites in the Upper Muna Field in Yakutsia. Along with pressure-temperature and oxygen fugacity calculations, these data are used to constrain the geotherm beneath this field and probe changes in chemical compositions with depth that could give rise to mantle layering. The data are further compared to those from neighbouring fields in order to reveal lateral differences in the thermal and compositional state of the underlying lithosphere. The authors also suggest links of the mantle structure to plumes and to diamond grade.
Overall, the paper presents an important dataset for a hitherto little-investigated mantle section of the Siberian craton, and as such this is a useful and welcome contribution. This may be suitable for Minerals journal, once some issues have been dealt with and in particular less supported hypotheses have either been strengthened or toned down. I hope that the comments and suggestions provided below and in the annotated file are useful for the authors to prepare a stronger manuscript.
Major issues:
- The writing is largely okay but in places is too succinct, as for some statements more support and evidence are needed. There are some lapses with respect to grammar, and in places the intended meaning is unclear. Instances are provided in the annotated manuscript, but I strongly recommend reading by a native speaker after the manuscript has been revised and prior to resubmission.
- The methods section is too short on detail. It should contain more basic information on the methods (not just references to earlier work) and in particular the quality control measures. For example, it would be good to indicate not just that some reference materials were measured as unknowns but also how close to accepted values the results are, which is a measure of accuracy.
- More attention should be paid to data quality. Major element compositions for some minerals appear to have been recalculated to 100%, which is not good practice. For some samples, data are reported where the sums are inacceptable. I suggest to remove all analyses with major element totals <98.5 or >101.5. Further, the trace element compositions reveal that kimberlitic material was probably co-ablated in some instances. This material can be present in small cracks that need not be optically visible when operating the laser. It is implausible to have 10s or 100s ppm Ba in garnet or cpx, given the very low distribution coefficients, even assuming very high Ba in a melt in equilibrium with these minerals. Luckily, this does not mean that the entire analysis needs to be discarded, as only elements with high kimberlite/mineral ratios are affected (Rb, Nb, Ta, Th and U in garnet and cpx, and additionally for Sr, the lightest REE and Pb in garnet). Please see Appdx G in Aulbach and Viljoen (2015 EPSL) where this has been treated in detail and quantitatively.
- The results section contains too much interpretation. The authors should take care to strictly separate the interpretation, which should be reserved for the discussion, from the purely descriptive part in the results. There are also a few instances in the introduction where it is unclear whether an interpretation is based on prior findings, in which case a reference is needed, or whether it reflects the authors’ opinions, in which case it should be removed to the discussion.
- A major issue with the manuscript is that it attempts to identify up 12 distinct mantle layers (according to Fig. 11) over a pressure interval of just 8 GPa. The identification of these layers is based on vague excursions and trends and may not hold up to statistical scrutiny, considering that some “layers” contain a low number of samples and if any realistic treatment of the uncertainties in pressure-temperature determinations employing 12 different formulations is included. I suggest to (1) provide information on the uncertainties associated with each of their geothermobarometer formulations, (2) to discuss whether these have been intercalibrated so that we can be sure that some of the purported trends and branches in the PT plot are significant and not due to a bias (e.g. if you apply the garnet and cpx single-mineral thermobarometers to garnet and cpx from xenoliths do you get the same answer or is there a systematic offset?), (3) to only discuss layers that are distinct outside these uncertainties and (4) bin their samples in 0.5 GPa intervals and provide averages or medians and standard deviations, which gives at least a basic statistical treatment. When I look at the PT plot I see that most samples fall on a conductive geotherm, that there is some scatter and that more samples in the deep than in the shallow lithosphere apparently fall on hotter geotherms, which could be explained by heating due to infiltration of metasomatic melts. Then the question becomes what are the geochemical signatures associated with melt infiltration. Maybe there are some additional significantly different intervals in there, but this could be better justified. All these purported layers also do not seem relevant to the discussion, where a sequence of events for formation and evolution of the lithosphere is established that recognises differences between shallow cold and deep hot parts and does not rely on additional “layers”.
- It is not clear that all the minerals employed to investigate lithospheric structure are derived from the lithospheric mantle rather than being crystallisation products of the host kimberlite melt (e.g. ilmenite), which would then amount to a discussion of mixed lithospheric and magmatic geotherms. Therefore, an early paragraph is useful to demonstrate that all have formed in the mantle prior to entrainment in the kimberlite. Moreover, the text frequently refers to dunitic or pyroxenitic types, and it is therefore important to first clarify what the criteria are for these distinctions (mineral compositional cut-offs?). Also, please introduce mineral abbreviations as early as possible and then use consistently. Remember that not all readers need have the necessary background to understand abbreviations.
- The authors repeatedly call for a subducted component, for which in my opinion there is no requirement or evidence in the data, where incompatible element enrichment is fully consistent with metasomatism by a melt from the carbonatite-kimberlite spectrum. Either way, the authors could provide either more references to the literature where similar patterns of enrichment have been demonstrated to be linked to a particular metasomatic agent, or provide some modelling. The same goes for the statement that peridotite melts mixed with metasome-derived melts. Can the authors provide some modelling to support this, or at least a more detailed discussion in support of this statement?
- The diamond formation scenario presented in the abstract and at the end of the discussion is too speculative. At the very least it requires a discussion of available age information, which for Siberian diamonds suggests old ages (summary in Gurney et al. 2010 Econ Geol + de Vries et al. 2013 GCA). It is also unclear why large diamonds should grow in equilibrium with a typically strongly oxidising melt, or why heating should be favourable to diamond growth. A scenario is proposed based on experiments whereby diamonds precipitate in the deepest part of the lithosphere at hot conditions, but just because this is viable does not mean that this happened. Is there any evidence from inclusions of the diamonds for formation at such conditions? At these temperatures N should be highly aggregated – any evidence for that? Moreover, in L58 the authors mention rounding and partial resorption of diamond during metasomatism, which seems more reasonable and is, of course, in contradiction to the growth scenario. There is a clear link between metasomatism by carbonated ultramafic melts and diamond destruction at various localities around the world (e.g. Aulbach et al. 2020 G3, Lithos). Why not give the discussion of large diamonds a different spin and explore what it took to preserve these despite evidence for a heating event and interaction with oxidising agents? For example, the heating and metasomatism occurred so close to eruption (based on strong compositional disequilibrium between garnet and cpx) that there was not enough time to destroy the diamond inventory. Maybe the many large diamonds are there because the smaller ones disappeared during heating/metasomatism/rounding?
- The supplementary files need more information. Please ensure that units are added to all sheets as necessary. The garnets virtually all have totals of 99.99, 100 or 100.01%, which suspiciously looks to me as if they were renormalised from “bad” totals. Please explain. Supp 1, line 98 contains “F” as the mineral – should be phlogopite based on composition? Supp 3 Deimos garnets has the header misaligned. When I add the totals there are several analyses which are inacceptable (<98.5 or >101.5 wt%) – these should be removed. Supp 3 Novinka cpx all have totals of exactly 100%, which suggests renormalisation. Please report unnormalised values and remove data that have unacceptable totals.
Please refer to the annotated manuscript, where comments or suggested changes are contained in speech bubbles that are placed at the lines which are concerned.

Author Response
Answer to Reviewer 1
- The writing is largely okay but in places is too succinct, as for some statements more support and evidence are needed. There are some lapses with respect to grammar, and in places the intended meaning is unclear. Instances are provided in the annotated manuscript, but I strongly recommend reading by a native speaker after the manuscript has been revised and prior to resubmission.
*I agree to some extent. Native speaker Prof. Hilary Downes have read and wrote “Thank you” Though possible that this is for content not for language.
- The methods section is too short on detail. It should contain more basic information on the methods (not just references to earlier work) and in particular the quality control measures. For example, it would be good to indicate not just that some reference materials were measured as unknowns but also how close to accepted values the results are, which is a measure of accuracy.
*Ok I provided the values and comparative diagram for the control samples.
More attention should be paid to data quality. Major element compositions for some minerals appear to have been recalculated to 100%, which is not good practice. For some samples, data are reported where the sums are inacceptable. I suggest to remove all analyses with major element totals <98.5 or >101.5.
*The data was received with Jeol 8320, and the Excel output gives an output in the normalized and non normalized values. Ok I’ll give non normalized
Further, the trace element compositions reveal that kimberlitic material was probably co-ablated in some instances.
*No we used clean unaltered grains without rims and visible inclusions
This material can be present in small cracks that need not be optically visible when operating the laser.
*I agree about Ba which sometimes in the fluid inclusions and it visible when they are decrepitated under the laser. But this time it was not observed
It is implausible to have 10s or 100s ppm Ba in garnet or cpx, given the very low distribution coefficients, even assuming very high Ba in a melt in equilibrium with these minerals.
*I see but the metasomatic minerals my incorporate some admixtures in the defects of structure which could not be determined optically
Luckily, this does not mean that the entire analysis needs to be discarded, as only elements with high kimberlite/mineral ratios are affected (Rb, Nb, Ta, Th and U in garnet and cpx, and additionally for Sr, the lightest REE and Pb in garnet).
*I checked again the primary data. Commonly in the samples with the high Ba have rather uniform counts. Though some are deviated I corrected these values.
Please see Appdx G in Aulbach and Viljoen (2015 EPSL) where this has been treated in detail and quantitatively.
*OK
The results section contains too much interpretation. The authors should take care to strictly separate the interpretation, which should be reserved for the discussion, from the purely descriptive part in the results. There are also a few instances in the introduction where it is unclear whether an interpretation is based on prior findings, in which case a reference is needed, or whether it reflects the authors’ opinions, in which case it should be removed to the discussion.
*Ok. Done
- A major issue with the manuscript is that it attempts to identify up 12 distinct mantle layers (according to Fig. 11) over a pressure interval of just 8 GPa. The identification of these layers is based on vague excursions and trends and may not hold up to statistical scrutiny, considering that some “layers” contain a low number of samples and if any realistic treatment of the uncertainties in pressure-temperature determinations employing 12 different formulations is included.
*This is not major item is not reconstruction of layering. Due to pervasive melt percolation the layering is smoothed. But the fluctuations of CaO in garnets and Fe – Mg show that layering existed.
I suggest to (1) provide information on the uncertainties associated with each of their geothermobarometer formulations,
(2) to discuss whether these have been intercalibrated so that we can be sure that some of the purported trends and branches in the PT plot are significant and not due to a bias (e.g. if you apply the garnet and cpx single-mineral thermobarometers to garnet and cpx from xenoliths do you get the same answer or is there a systematic offset?), (3) to only discuss layers that are distinct outside these uncertainties and (4) bin their samples in 0.5 GPa intervals and provide averages or medians and standard deviations, which gives at least a basic statistical treatment.
*The details of the calibration of the thermobarometers you can find in previous papers (Ashchepkov et al., 2010; 2014 ;2017). Commonly the precision is around 2 -5 kbars For the clinopyroxene method it is better and is near 2kbat and method is comparable to (Nimis, Taylor, 2000) But it works for Peridotitic, basaltic and eclogitic associations together
When I look at the PT plot I see that most samples fall on a conductive geotherm, that there is some scatter and that more samples in the deep than in the shallow lithosphere apparently fall on hotter geotherms, which could be explained by heating due to infiltration of metasomatic melts.
*Yes you are correct And the continuations of the geotherms are subaduabatic which is crossing conductive geotherms. And this is common for most mantle geotherms (see Ashchepkov et al., 2019).
Then the question becomes what are the geochemical signatures associated with melt infiltration. Maybe there are some additional significantly different intervals in there, but this could be better justified. All these purported layers also do not seem relevant to the discussion, where a sequence of events for formation and evolution of the lithosphere is established that recognises differences between shallow cold and deep hot parts and does not rely on additional “layers”.
*There are differences of course in geochemistry between the protokimberlite related minerals – mainly Cpx ,Ilm and garnets which are not equilibrated in xenoliths. We analyzed Cpx from the xenoliths which nearly completely refer to refertilized type often with the Th anomalies source but garnets mainly have common peridotitic geochemistry
- It is not clear that all the minerals employed to investigate lithospheric structure are derived from the lithospheric mantle rather than being crystallisation products of the host kimberlite melt (e.g. ilmenite), which would then amount to a discussion of mixed lithospheric and magmatic geotherms. Therefore, an early paragraph is useful to demonstrate that all have formed in the mantle prior to entrainment in the kimberlite.
*Of course the protokimberlitic stage may differ from the kimberlite eruption in million years
Moreover, the text frequently refers to dunitic or pyroxenitic types, and it is therefore important to first clarify what the criteria are for these distinctions (mineral compositional cut-offs?).
*The dunitic garnets are all sub-calcic and they all have S type REE patterns
Also, please introduce mineral abbreviations as early as possible and then use consistently. Remember that not all readers need have the necessary background to understand abbreviations.
*Ok
- The authors repeatedly call for a subducted component, for which in my opinion there is no requirement or evidence in the data, where incompatible element enrichment is fully consistent with metasomatism by a melt from the carbonatite-kimberlite spectrum.
*The high U, Ba, Sr, LILE have mainly the relations to the subduction stage, though some carbonatitic melts also
Either way, the authors could provide either more references to the literature where similar patterns of enrichment have been demonstrated to be linked to a particular metasomatic agent, or provide some modelling. The same goes for the statement that peridotite melts mixed with metasome-derived melts. Can the authors provide some modelling to support this, or at least a more detailed discussion in support of this statement?
*It is not easy to model because nobody have seen protokimberlites as well as subducted related fluids. I reconstructed with the KD protokimberlites (Ashchepkov et al.,) protokimberlites. To make model with AFC the deal of several ours but probably not for this paper because it already contain a lot of material
- The diamond formation scenario presented in the abstract and at the end of the discussion is too speculative. At the very least it requires a discussion of available age information, which for Siberian diamonds suggests old ages (summary in Gurney et al. 2010 Econ Geol + de Vries et al. 2013 GCA).
*These are Devonian kimberlites only I added 2 references. Precambrian ages are suggested by V.Afanasiev and colleagues for Kutungde river placers. There some estimates about 420-400 MA for the Toluopskoe field in the North and probably the nearby field
It is also unclear why large diamonds should grow in equilibrium with a typically strongly oxidising melt, or why heating should be favourable to diamond growth. A scenario is proposed based on experiments whereby diamonds precipitate in the deepest part of the lithosphere at hot conditions, but just because this is viable does not mean that this happened. Is there any evidence from inclusions of the diamonds for formation at such conditions?
*Look please in the P- FO2 diagrams Most of the Cpx and even ilmenites from Zapolyarnaya, Komsomolskaya-Magnitnaya are inside the diamond stability field which is rather uncommon feature of the protokimberlitic melts They were rapidly reduced after the intrusions probably due to the contamination And thus may be the source of the reduced carbon for diamonds
At these temperatures N should be highly aggregated – any evidence for that? Moreover, in L58 the authors mention rounding and partial resorption of diamond during metasomatism, which seems more reasonable and is, of course, in contradiction to the growth scenario.
It is still anclear till somebody dated diamond inclusions or diamonds / The examples are known
*in Ebelyakh they gave Devonian ages for rutile (Ragozin) inclusions in diamonds and in Ural also (Laiginhas et al., 2009).
There is a clear link between metasomatism by carbonated ultramafic melts and diamond destruction at various localities around the world (e.g. Aulbach et al. 2020 G3, Lithos). Why not give the discussion of large diamonds a different spin and explore what it took to preserve these despite evidence for a heating event and interaction with oxidising agents? For example, the heating and metasomatism occurred so close to eruption (based on strong compositional disequilibrium between garnet and cpx) that there was not enough time to destroy the diamond inventory. Maybe the many large diamonds are there because the smaller ones disappeared during heating/metasomatism/rounding?
*I suggest both mechanism exists.
Look at the diamonds images. Large diamonds are well shaped without the sigh of the dissolution. And many of light diamonds also. Dark diamond are more rounded. They should be ancient corresponding possibly to the subduction related malts
The diamonds from Zapolyarnaya differ in their characteristics from the other regions in. “Zapolyarnaya, pipes (about 100 determinations) are characterized by a decrease in the average δ13C (−4.59, −4.50,−4.04‰) and by relatively low average nitrogen contents (93, 254, 304 ppm, respectively) which probably related to the crystallization by the fluids. The pyrolysis of ethane—C2H6 → CH4 + H2 + Cdiam—is assumed to be a model of diamond precipitation from fluid” (Ukhanov et al.,2011)
- The supplementary files need more information. Please ensure that units are added to all sheets as necessary. The garnets virtually all have totals of 99.99, 100 or 100.01%, which suspiciously looks to me as if they were renormalised from “bad” totals. Please explain. Supp 1, line 98 contains “F” as the mineral – should be phlogopite based on composition? Supp 3 Deimos garnets has the header misaligned. When I add the totals there are several analyses which are inacceptable (<98.5 or >101.5 wt%) – these should be removed.
*Ok I corrected the supplementary files added the results of control samples.
Supp 3 Novinka cpx all have totals of exactly 100%, which suggests renormalisation. Pleasereport unnormalised values and remove data that have unacceptable totals.
*Normalized because there is Excel output from Jeol 8320 – Ok I substituted.
Please refer to the annotated manuscript, where comments or suggested changes are contained in speech bubbles that are placed at the lines which are concerned.
Comment L31-35 Please discuss under which circumstances all these different PT branches (at least six listed here) have geological meaning, rather than arising from inaccuracies due to the use of different geothermobarometers that are not perfectly inter-calibrated/comparable. I think when you acknolwedge the uncertainties, there would be no more than a conductive geotherm with scatter due to isobaring melt-advected heating plus perhaps a deep kink also reflecting late heating. Why would different minerals consistently equilibrate to separate geothermal gradients? Also, ilmenite is rarely reported as a primary mineral in peridotites, but is a typical megacryst mineral, and the origin of those is still debated. See also major comment
*The thermobarometesr are internally consistent . I’ll send to you the file for Udachnaya mantle xenoliths where the mineral are more equilibrated (excluding sheared perifotites)
garnet geotherms are commonly more low temperature than those for pyroxenes. Thoghh it depends
More ancient commonly Ilmenites are often among the metasomatic veins but it seem that no all them associated with the protokimberlites In Sytykanskaya pipe the associated phlogopites gives the ages to 1.5 Ga. The Garnets and Cpx are geochemically different in Zapolyarnaya.
Thank you very much for the useful comments and detail work on the manuscript.
Nobody gives the scale of precision on the PT diagrams I should say that many barometers are not very precise Ryan, 1996 and Grutter, 2004 have the everage error abou15- 20 kbars.
Elevated concentrations of the listed elements can be satisfied by interaction with kimberlite-like melts alone. This is not evidence for involvement of subducted material, which would be Nb-poor
*Kimberlites practically do not react with the mantle xenoliths, may dissolved the diffusion profiles exist only in olivine mush in the kimberlites. The deep seated pyroxene could better dissolved than reacted. The problem of Nb exists. Rutile is the mineral which has the Nb KD much higher than ta and divide them in the geochemical processes. This mineral is typical for subduction related processes and especially for eclogites. The magmas melting such eclogites and precipitating in mantle could produce such high Nb metasomatism better than for kimberlites and the metasomatism of such melts could also lead to the Nb enrichment
Thank you very much dor the detail review which is very useful. Best wishes Igor Ashchepkov
Reviewer 2 Report
This is a very thorough study by Ashchepkov and coauthors on mantle xenoliths from the Zapolyarnaya kimberlite pipe. The authors reported a new set of geochemical data on the mineral compositions. Applying single-mineral geothermometers and geobarometers, they constructed the geothermal and fO2 profiles of continental lithosphere mantle beneath the kimberlite pipe. Further, they proposed a synthesized geological interpretation regarding the metasomatism processes in the lithosphere mantle, likely involving a hot mantle plume and multi-stage melt percolation. In addition, the authors also discussed the potential implications of their study on diamond formation. Overall, this paper is well written and meets the scope of Minerals. I think it should be published at Minerals after some minor revisions. Following are my comments that may help the authors improve their manuscript.
1. As the authors showed in the paper, major and trace element compositions of mantle minerals show considerable variations. These chemical variations involve at least two distinct processes: (1) magma crystallization or melt-rock reaction; (2) subsolidus re-equilibration in the lithosphere mantle. Because of the subsolidus re-equilibration, petrologists could use mineral-based geothermometers and geobarometers to constrain the lithosphere geotherms using either major or trace elements in minerals. To assess the magmatic processes prior to subsolidus re-equilibration, it is important to identify chemical records that are insensitive to subsolidus re-equilibration. Often, trace elements (e.g., REE) are used for this purpose. However, Sun and Liang (2014) showed that this assumption may not be always true. With this being said, I suggest that in the paper the authors should comment on the potential effects of subsolidus re-equilibration on the trace element compositions of mantle minerals and further the estimated parental melt compositions (see their Figs. 12-15).
- Sun, C. and Liang, Y., 2014. An assessment of subsolidus re-equilibration on REE distribution among mantle minerals olivine, orthopyroxene, clinopyroxene, and garnet in peridotites. Chemical Geology, 372, pp.80-91.
2. Lines 393-394: The authors presented a very interesting observation here, i.e., "Spreading pyroxenitic and dunite-harzburgitic arrays to the bottom of the SCLM". They interpreted that this observation suggests "additional differentiation carbonation or hydration to cross the dense SCLM." (Lines 400–401). In fact, there are recent experimental works on the reactions between photo-kimberlite melts and deeper lithosphere mantle. The new observations from this manuscript appear to be in excellent agreement with the experimental results of Sun and Dasgupta (2019), which showed that reactions between CO2-rich melts and peridotites would generate garnet-bearing wehrlite or pyroxenite.
- Sun, C. and Dasgupta, R., 2019. Slab–mantle interaction, carbon transport, and kimberlite generation in the deep upper mantle. Earth and Planetary Science Letters, 506, pp.38-52.
3. Figures 10-11: No labels for the x-axis. fO2 relative to QFM?
4. "FO2" appears in many places of the manuscript. Usually it is writen as "fO2".
5. Line 423: "super plums" should read as "super plumes".
6. Lines 441-443: Slab-released fluids are generally depleted in Zr and other HFSE, right?
7. Figure 10 caption: "It addition" should read as "Its addition"
Author Response
Answer to Reviewer 2
This is a very thorough study by Ashchepkov and coauthors on mantle xenoliths from the Zapolyarnaya kimberlite pipe. The authors reported a new set of geochemical data on the mineral compositions. Applying single-mineral geothermometers and geobarometers, they constructed the geothermal and fO2 profiles of continental lithosphere mantle beneath the kimberlite pipe. Further, they proposed a synthesized geological interpretation regarding the metasomatism processes in the lithosphere mantle, likely involving a hot mantle plume and multi-stage melt percolation. In addition, the authors also discussed the potential implications of their study on diamond formation. Overall, this paper is well written and meets the scope of Minerals. I think it should be published at Minerals after some minor revisions. Following are my comments that may help the authors improve their manuscript.
Thank you
1. As the authors showed in the paper, major and trace element compositions of mantle minerals show considerable variations. These chemical variations involve at least two distinct processes: (1) magma crystallization or melt-rock reaction; (2) subsolidus re-equilibration in the lithosphere mantle. Because of the subsolidus re-equilibration, petrologists could use mineral-based geothermometers and geobarometers to constrain the lithosphere geotherms using either major or trace elements in minerals. To assess the magmatic processes prior to subsolidus re-equilibration, it is important to identify chemical records that are insensitive to subsolidus re-equilibration. Often, trace elements (e.g., REE) are used for this purpose. However, Sun and Liang (2014) showed that this assumption may not be always true. With this being said, I suggest that in the paper the authors should comment on the potential effects of subsolidus re-equilibration on the trace element compositions of mantle minerals and further the estimated parental melt compositions (see their Figs. 12-15).
*Thank you Yes I knew this, commonly the Cr content of the pyrope garnets have negative correlation with the La/Ybn pressures and temperatures.
- Sun, C. and Liang, Y., 2014. An assessment of subsolidus re-equilibration on REE distribution among mantle minerals olivine, orthopyroxene, clinopyroxene, and garnet in peridotites. Chemical Geology, 372, pp.80-91.
- Lines 393-394: The authors presented a very interesting observation here, i.e., "Spreading pyroxenitic and dunite-harzburgitic arrays to the bottom of the SCLM". They interpreted that this observation suggests "additional differentiation carbonation or hydration to cross the dense SCLM." (Lines 400–401). In fact, there are recent experimental works on the reactions between photo-kimberlite melts and deeper lithosphere mantle. The new observations from this manuscript appear to be in excellent agreement with the experimental results of Sun and Dasgupta (2019), which showed that reactions between CO2-rich melts and peridotites would generate garnet-bearing wehrlite or pyroxenite.
- Sun, C. and Dasgupta, R., 2019. Slab–mantle interaction, carbon transport, and kimberlite generation in the deep upper mantle. Earth and Planetary Science Letters, 506, pp.38-52.
*The spreading of the Ca and sub ca array is common feature of the Lithosphereic mantle It supports the idea that the mantle lithosphere was melted after the SCLM growth (Herzberg , 2004; Ionov et al., 2010) And the differentiation probably tok place after this melting.
- Figures 10-11: No labels for the x-axis. fO2 relative to QFM?
*Yes
- "FO2" appears in many places of the manuscript. Usually it is written as "fO2".
*For the name of diagram I do not use lower cases just to mark Yes this is ∆LogFo2 relative quartz fayalite magnetite (QMF) buffer I wrote in text
- Line 423: "super plums" should read as "super plumes".
*Ok
- Lines 441-443: Slab-released fluids are generally depleted in Zr and other HFSE, right?
*I am not sure. Opposite I have seen and analyzed the zircon Phl –Cpx bearing veins in mantle peridotites in Russia far East. In Fineiro and other orogenic massifs all zircons are in the phlogopite veins H2O fluids are associated with phlogopites also
- Figure 10 caption: "It addition" should read as "Its addition"
*Changed
This contribution presents detailed mineral chemistry and thermobarometric data for heavy mineral separates and xenolith minerals from Yakutian kimberlites. These are used to comment upon the structure of the lithospheric mantle and to infer some details regarding diamond prospectivity in this region. The volume and quality of data alone warrant publication, and their utility to future students of kimberlites and of the mantle are clear. However, I must suggest some improvements towards the writing and presentation, which are listed below, and which probably sum to something between moderate and major revisions.
*Thank you very much for the review which is very useful.
Reviewer 3 Report
This contribution presents detailed mineral chemistry and thermobarometric data for heavy mineral separates and xenolith minerals from Yakutian kimberlites. These are used to comment upon the structure of the lithospheric mantle and to infer some details regarding diamond prospectivity in this region. The volume and quality of data alone warrant publication, and their utility to future students of kimberlites and of the mantle are clear. However, I must suggest some improvements towards the writing and presentation, which are listed below, and which probably sum to something between moderate and major revisions.
Major comments
- From the information provided in the introduction, the purpose of the study can be inferred with some guesswork, but should be stated much more explicitly. What I mean is, you state clearly what you did, but not necessarily why you did it. At the end of the introduction, I recommend a clearer statement to accompany lines 65-74, including (but not limited to) the following reasons that I can see from reading this manuscript: better understanding of the lithospheric structure and composition, and understanding effects of kimberlite metasomatism upon diamond grades in this location. This information should also probably be featured somewhere in the abstract.
- Whereas the obvious quality and amount of data comes through strongly, the interpretations in the discussion section are not, in my opinion, sufficiently supported. Just to be clear, I’m not necessarily questioning the outcomes of the study, but it is not obvious how these outcomes have been reached, as some interpretations are made without a great deal of supporting evidence. I think even just a few sentences backed up by relevant citations, mostly in order to lead a reader through the authors’ logical arguments step by step, would be enough to make up for this shortfall. Just as one example, lines 431-450 contain a lot of statements about the state of the lithospheric mantle, but provide precious little supporting evidence, either from the figures and data from this study or from previous studies.
- Section 5 (lines 159-219), which features the description of mineral chemical data, is currently not very well written at all. I admit that I often have trouble with results sections myself, but I find these to be among the most important sections in papers as, if nothing else, papers with high quality data will always be useful to future studies. I implore the authors to consider a thorough rethink of the key information that needs to be presented, and for this information to be provided in as systematic and clear a fashion as possible. A couple of examples that I think must be examined (although please look through in detail!):
- Lines 180-181 currently retain very little meaning, and in mentioning a figure of 6 wt.% FeO, directly contradict a previous statement that FeO varies from 1-5 wt.%
- Terms FeO-poor and FeO-rich are not defined by any particular value interval, and I suggest that if this is to be retained, value ranges be defined in this section (for other parameters as well, if desired)
- Section 6 (lines 220-295) is a little confusing. As you will see below, I have a few specific examples of things that are unclear, but indeed this section is full of references to trends that are not clear or obvious on any of the relevant figures. I recommend perhaps labelling said trends on the figures, for example, or at least adding lines/fields to delineate these trends that can be described in the figure captions. Although this detailed thermobarometry is not my deepest area of expertise, it seems that if there are trends on these figures, they are more closely related to the thermobarometer that has been employed? I wonder if the authors could perhaps comment on this in this section as well, as if this is not the case (which it may well not be!) then the symbols and so on that have been employed must be explained better.
- Section 7 and the plots with trace element patterns could benefit from more comparisons with data, even if such comparisons would draw from other localities outside of Russia. I would certainly recommend comparisons at least to works that have examined kimberlitic metasomatism (for example, Grégoire et al. 2002, 2003, spring to mind). As well as this being a useful point of reference, it would also add a great deal of weight and importance to this work that would link common kimberlitic metasomatic signatures in all global kimberlite occurrences.
- I appreciate that English may not be the corresponding author’s first language, and certainly in some sections the language usage is quite good, but there are other sections that really must be improved in order to aid readers’ understanding (e.g., sections 8.2, 8.3, and 8.4 to make your preferred model clearer; sorry that I don’t have many scientific comments for this section at the moment). Once all other changes have been made, please do go through carefully and make sure that everything is as clear as possible. This will vary from the simplest of typos (a few of which are listed below) to more comprehensive rewriting of some sentences.
Minor comments
Line 4: I think this should be Yakutia, not Yukutia
Line 19: I think instead of EPMS you mean EPMA
Line 20: I recommend changing sub-Ca to sub-calcic, both here and in other instances throughout
Line 22-24: I don’t think the meaning of this sentence is clear. Do you mean that the “PT estimates…fall along the convective branch of the geotherm”? I also suggest rewriting the end of this sentence to clarify the meaning of “inflected at 6 GPa which is connected with the ilmenite PTX trend.”
Line 26-31: on line 29, “for the dunitic garnets” comes a bit out of nowhere, as at first glance the way this section is written makes it seem like the trace element characteristics you have identified are a continuation of a description of the clinopyroxene groupings in the previous sentence. If the sentences are separate, i.e. one about clinopyroxene and one about garnet, I recommend rearranging the second sentence, i.e. “Dunitic garnets of the lower pressure groups have elevated …”
Line 34: Are the cpx-phl veinlets related to kimberlitic metasomatism?
Line 35: the relevance of “there are also separate pyroxenitic branches” is not clear in this context, and I suggest either removing from the abstract or a slight elaboration.
Line 36: please change “alkalis” to “alkali metals”
Line 46-47: Unless this is supposed to be a general statement, broadly about diamond exploration, please clarify, either with a reference (a commercial report, perhaps?) or with a specific example, what these difficulties are in this location. Alternatively, if this is supposed to be a general statement, perhaps you might consider moving this sentence to the very start, with the new meaning being “it is difficult to find diamond exploration targets, but the Upper Muna kimberlite field is quite promising”
Line 57: Please consider adding a specific website link (to the ALROSA reports) so that readers can easily find this information.
Line 58: please change “clean” to something with the equivalent intended meaning (e.g., (relatively) inclusion-free?)
Line 60: I suggest changing “the field” to “the Upper Muna kimberlite field”, just to be clear
Line 68-70: do you mean that the clinopyroxene data presented by Dymshits do not form a continuous geotherm?
Line 78: please change “Sr-Nd dating” to either “Rb-Sr and Sm-Nd dating” or just “Sm-Nd dating”, assuming that this is the intended meaning
Line 80: I’m not sure that many people would be comfortable with the term “porphyritic kimberlites”; I suggest changing this to “macrocrystic kimberlites”
Line 80-81: please add values for MgO, Al2O3, CO2 contents if available
Line 82: is this HIMU field trend supported by Pb isotope data, if they exist?
Lines 81-83: Are these isotopic data from bulk-rock samples, or mineral separates (e.g., perovskite)? Furthermore, can the CO2 content of these samples be related to the amount of macrocrysts in those samples, and can this relationship rather be related to the degree of contamination?
Line 83: I would suggest moving this statement about Mg-rich samples up, above the isotopic data section. `
Line 85: please clarify what (reference data) was used to normalise the La/Ybn ratio, and also consider rewriting as either Lan/Ybn or (La/Yb)n
Line 94: please provide more detail in the caption for Figure 2, regarding the symbols used in this figure.
Line 96: this caption is missing a label for panel A showing a photo of the open-pit mine at Zapolyarnaya.
Line 104: suggest “…although they are present in all other bodies.”
Line 112: I’m not familiar with the term “nests” here. Can you please elaborate and/or provide an image of this?
Line 115: can you use a different word to “scattered” here? The meaning of this word in this sentence is currently not clear.
Line 122: Fig. 4, panel D, whereas in the other panels I can see what you are referring to without any figure labels, it’s not clear what is what in panel D, and I would recommend labels in the figure in addition to the current caption.
Line 123: suggest “Pale olivine aggregates resemble those found in polymict breccias…”
Lines 138-140: I’m not sure I completely understand why two different procedures were used to analyse xenolith minerals vs KIM. Is this just because the machines were upgraded in between the two analytical runs? If this is the reason, then fine; if this is not the reason, I suggest that more explanation might be required.
Line 151-152: I suggest also including a statement about the accuracy of analyses of these standard reference materials here, relative to published/accepted values.
Lines 163-168, and Fig. 5: the main focus of this description appears to be about the CaO-Cr2O3 diagram, and at the moment it’s not clear, due to the small size of these panels, what the sub-trends are that you’re referring to. I wonder if it would be helpful to reduce this figure to only four panels of CaO-Cr2O3 plots (one panel per locality), so that these can be bigger for readers to better understand what’s going on? The other panels could perhaps be removed to a supplementary figure. I am also not particularly convinced as to whether there’s any statistical significance of your statement regarding higher TiO2 at low Cr2O3, given the appearance of your data on these plots as relatively broad clouds. The final statement in this paragraph, regarding Na2O enrichment, is also unclear and needs to be rewritten.
In Fig. 5, I’m not sure what the abbreviation AKB (in the key) is referring to. Why also are there some green symbols on the Zapolyarnaya plots; does this relate to the preliminary analyses referred to on lines 131-135? Finally, although many readers could make an educated guess at this, what are the meanings of the grey field and bar graphs in the CaO-Cr2O3 plots in Fig. 5? All of this information should be included in the figure caption.
Lines 176-177: similar to the previous comment, I’m not sufficiently convinced by the current Fig. 5 to be able to say that there is relatively more Ti-Na metasomatism in the garnets from Novinka or Komsomolskaya-Magnitnaya, relative to the garnets from the other localities.
Line 179-180: “close to omphacites” is ambiguous, as you could be referring to either location (i.e. implying that omphacites are present in your samples) or, as I think you mean, that these compositions trend towards the ideal composition of omphacite. Therefore, please consider using a different expression here.
Fig 8, lines 210-215: I wonder if there are more appropriate plots to show chromite compositional variability, for example those in Roeder and Schulze (2008, JPetrol). For example, in the plots provided, I’m not sure the term “ulvospinel branch” (line 212) carries much meaning, because it’s hard to relate the data to other potentially overlapping compositions in 2-D space.
Line 214: I think this statement that ulvospinels were more oxidised rather better belongs in a discussion section, or else should be backed up in this instance with a bit more info and/or a relevant citation for comparison.
Figure 10: A comment particularly for this figure, but also generally for many other figures, is that in a printed version of this manuscript, a lot of the symbols on figures would be rather small and difficult to identify. I appreciate the detail that has been put into some figure captions, such as this one, but I think if the figure and hence the symbols could be enlarged, many readers would appreciate the effort this might take. At the same time, I’m curious as to why Fe#, and not Mg#, is frequently used in this draft. Can the authors give an explanation for this?
Line 231: do you mean that “the garnet geotherm indicates a (relatively) high geothermal gradient”?
Line 256: I’m not sure what is meant by ilmenite trend here, perhaps if possible this could be explained in a bit more detail here? In fact, I think a bit more detail overall is required for all the trends identified in lines 255-263.
Line 271-273: This sentence appears to be somewhat contradictory. Can the authors perhaps rephrase?
Line 380-383: I think “they” here needs to be replaced with something more specific, to make it clear that you are discussing either amphibole generally, or amphiboles with a wide compositional range.
Line 384-385: I’m not certain that phlogopite addition, per se, can be used specifically as a sign of subduction-related metasomatism, and I would either recommend a different citation to the one listed, or else provide a further or more specific statement as to the origins of phlogopite within the SCLM.
Line 407-408: what does “crust granitization” mean here?
Line 426: I think here you should be referring to a figure other than Fig. 6?
Lines 434-450: Please provide more information regarding your interpretation of the relative order of the metasomatic events. At the moment it is not clear how you have reached these conclusions.
Line 462-465: is it possible to qualify how HFSE were lost (e.g., ilmenite accumulation?) and maybe even quantify the extent to which this might have occurred? Could this be related to (ilmenite) megacryst formation?
Line 467: What is the source of this “further enrichment”?
Line 473: I would recommend instead showing a plot of gt/cpx partition coefficients, compared against published data for different settings.

Author Response
Answer to Reviewer 3
Major comments
- From the information provided in the introduction, the purpose of the study can be inferred with some guesswork, but should be stated much more explicitly. What I mean is, you state clearly what you did, but not necessarily why you did it. At the end of the introduction, I recommend a clearer statement to accompany lines 65-74, including (but not limited to) the following reasons that I can see from reading this manuscript: better understanding of the lithospheric structure and composition, and understanding effects of kimberlite metasomatism upon diamond grades in this location. This information should also probably be featured somewhere in the abstract.
*I added two paragraphs with the porposes
- Whereas the obvious quality and amount of data comes through strongly, the interpretations in the discussion section are not, in my opinion, sufficiently supported. Just to be clear, I’m not necessarily questioning the outcomes of the study, but it is not obvious how these outcomes have been reached, as some interpretations are made without a great deal of supporting evidence. I think even just a few sentences backed up by relevant citations, mostly in order to lead a reader through the authors’ logical arguments step by step, would be enough to make up for this shortfall. Just as one example, lines 431-450 contain a lot of statements about the state of the lithospheric mantle, but provide precious little supporting evidence, either from the figures and data from this study or from previous studies. *I seriously changed the discussion
- Section 5 (lines 159-219), which features the description of mineral chemical data, is currently not very well written at all. I admit that I often have trouble with results sections myself, but I find these to be among the most important sections in papers as, if nothing else, papers with high quality data will alwaysbe useful to future studies. I implore the authors to consider a thorough rethink of the key information that needs to be presented, and for this information to be provided in as systematic and clear a fashion as possible. A couple of examples that I think must be examined (although please look through in detail!):
- Lines 180-181 currently retain very little meaning, and in mentioning a figure of 6 wt.% FeO, directly contradict a previous statement that FeO varies from 1-5 wt.%
- Terms FeO-poor and FeO-rich are not defined by any particular value interval, and I suggest that if this is to be retained, value ranges be defined in this section (for other parameters as well, if desired)
*Changed
- Section 6 (lines 220-295) is a little confusing. As you will see below, I have a few specific examples of things that are unclear, but indeed this section is full of references to trends that are not clear or obvious on any of the relevant figures. I recommend perhaps labelling said trends on the figures, for example, or at least adding lines/fields to delineate these trends that can be described in the figure captions.
- Although this detailed thermobarometry is not my deepest area of expertise, it seems that if there are trends on these figures, they are more closely related to the thermobarometer that has been employed?
*The thermobarometers are internally consistent
I wonder if the authors could perhaps comment on this in this section as well, as if this is not the case (which it may well not be!) then the symbols and so on that have been employed must be explained better.
*Some trends were added to fig. 10.
- Section 7 and the plots with trace element patterns could benefit from more comparisons with data, even if such comparisons would draw from other localities outside of Russia. I would certainly recommend comparisons at leastto works that have examined kimberlitic metasomatism (for example, Grégoire et al. 2002, 2003, spring to mind). As well as this being a useful point of reference, it would also add a great deal of weight and importance to this work that would link common kimberlitic metasomatic signatures in all global kimberlite occurrences.
There is a compilation of such diagrams for garnets in Ashchepkov at all, 2019.(see Research gate) He compilation of the worldwide data – the long story Possibly in Mains in Rock data base it already done. I added the references for the comparison in discussion
- I appreciate that English may not be the corresponding author’s first language, and certainly in some sections the language usage is quite good, but there are other sections that really must be improved in order to aid readers’ understanding (e.g., sections 8.2, 8.3, and 8.4 to make your preferred model clearer; sorry that I don’t have many scientific comments for this section at the moment). Once all other changes have been made, please do go through carefully and make sure that everything is as clear as possible. This will vary from the simplest of typos (a few of which are listed below) to more comprehensive rewriting of some sentences.
I corrected when it was possible. Prof Hilary owns also worked on this
Minor comments
Line 4: I think this should be Yakutia, not Yukutia
Ok
Line 19: I think instead of EPMS you mean EPMA
Ok
Line 20: I recommend changing sub-Ca to sub-calcic, both here and in other instances throughout
Ok
Line 22-24: I don’t think the meaning of this sentence is clear. Do you mean that the “PT estimates…fall along the convective branch of the geotherm”? I also suggest rewriting the end of this sentence to clarify the meaning of “inflected at 6 GPa which is connected with the ilmenite PTX trend.”
*Ok corrected
Line 26-31: on line 29, “for the dunitic garnets” comes a bit out of nowhere, as at first glance the way this section is written makes it seem like the trace element characteristics you have identified are a continuation of a description of the clinopyroxene groupings in the previous sentence. If the sentences are separate, i.e. one about clinopyroxene and one about garnet, I recommend rearranging the second sentence, i.e. “Dunitic garnets of the lower pressure groups have elevated …”
*Ok rearranged
Line 34: Are the cpx-phl veinlets related to kimberlitic metasomatism?
*Probably initially not But may be reactivated.
Line 35: the relevance of “there are also separate pyroxenitic branches” is not clear in this context, and I suggest either removing from the abstract or a slight elaboration.
*changed
Line 36: please change “alkalis” to “alkali metals”
*Ok
Line 46-47: Unless this is supposed to be a general statement, broadly about diamond exploration, please clarify, either with a reference (a commercial report, perhaps?) or with a specific example, what these difficulties are in this location. Alternatively, if this is supposed to be a general statement, perhaps you might consider moving this sentence to the very start, with the new meaning being “it is difficult to find
*I added reference
Line 57: Please consider adding a specific website link (to the ALROSA reports) so that readers can easily find this information.
diamond exploration targets, but the Upper Muna kimberlite field is quite promising”
http://www.alrosa.ru/верхне-мунское-месторождение-алроса-2/
https://ysia.ru/alrosa-dobyla-na-verhne-munskom-mestorozhdenii-pervyj-s-nachala-ego-otrabotki-krupnyj-yarkij-almaz/
https://www.interfax.ru/business/235642
Line 58: please change “clean” to something with the equivalent intended meaning (e.g., (relatively) inclusion-free?)
*removed
Line 60: I suggest changing “the field” to “the Upper Muna kimberlite field”, just to be clear
*Ok
Line 68-70: do you mean that the clinopyroxene data presented by Dymshits do not form a continuous geotherm?
*Yes it form continuous geotherm but with groups and clots
Line 78: please change “Sr-Nd dating” to either “Rb-Sr and Sm-Nd dating” or just “Sm-Nd dating”, assuming that this is the intended meaning
*Ok
Line 80: I’m not sure that many people would be comfortable with the term “porphyritic kimberlites”; I suggest changing this to “macrocrystic kimberlites”
*ok
Line 80-81: please add values for MgO, Al2O3, CO2 contents if available
*Ok
Line 82: is this HIMU field trend supported by Pb isotope data, if they exist?
*No
Lines 81-83: Are these isotopic data from bulk-rock samples, or mineral separates (e.g., perovskite)? Furthermore, can the CO2 content of these samples be related to the amount of macrocrysts in those samples, and can this relationship rather be related to the degree of contamination?
*These data were published by Kostrovitsky et al., 2007 and Sun et al., 2014
Line 83: I would suggest moving this statement about Mg-rich samples up, above the isotopic data section.
*Ok
Line 85: please clarify what (reference data) was used to normalise the La/Ybn ratio, and also consider rewriting as either Lan/Ybn or (La/Yb)n
*Ok
Line 94: please provide more detail in the caption for Figure 2, regarding the symbols used in this figure.
*Ok
Line 96: this caption is missing a label for panel A showing a photo of the open-pit mine at Zapolyarnaya.
Line 104: suggest “…although they are present in all other bodies.”
Line 112: I’m not familiar with the term “nests” here. Can you please elaborate and/or provide an image of this?
*Removed’. This is common for all peridotites I ‘ll show next time simply necessary to polish samples
Line 115: can you use a different word to “scattered” here? The meaning of this word in this sentence is currently not clear.
*changed
Line 122: Fig. 4, panel D, whereas in the other panels I can see what you are referring to without any figure labels, it’s not clear what is what in panel D, and I would recommend labels in the figure in addition to the current caption.
Line 123: suggest “Pale olivine aggregates resemble those found in polymict breccias…”
Ok I added the reference
Lines 138-140: I’m not sure I completely understand why two different procedures were used to analyse xenolith minerals vs KIM. Is this just because the machines were upgraded in between the two analytical runs? If this is the reason, then fine; if this is not the reason, I suggest that more explanation might be required.
*I analyzed minerals at first in IGM SB RAS than they stopped laser ablation for minerals and now I work in IIC SB RAS
Line 151-152: I suggest also including a statement about the accuracy of analyses of these standard reference materials here, relative to published/accepted values.
*I Made a sheet in Supplement2- Control and diagrams for the comparison There are systematic differences for Left part because in solution the material was leached removing all admixtures which mechanically enter in the ablate material and in solutions it seems some HFSE are systematically lower
Lines 163-168, and Fig. 5: the main focus of this description appears to be about the CaO-Cr2O3 diagram, and at the moment it’s not clear, due to the small size of these panels, what the sub-trends are that you’re referring to. I wonder if it would be helpful to reduce this figure to only four panels of CaO-Cr2O3 plots (one panel per locality), so that these can be bigger for readers to better understand what’s going on? The other panels could perhaps be removed to a supplementary figure. I am also not particularly convinced as to whether there’s any statistical significance of your statement regarding higher TiO2 at low Cr2O3, given the appearance of your data on these plots as relatively broad clouds. The final statement in this paragraph, regarding Na2O enrichment, is also unclear and needs to be rewritten.
*Ok Figure changed and text also
In Fig. 5, I’m not sure what the abbreviation AKB (in the key) is referring to. Why also are there some green symbols on the Zapolyarnaya plots; does this relate to the preliminary analyses referred to on lines 131-135? Finally, although many readers could make an educated guess at this, what are the meanings of the grey field and bar graphs in the CaO-Cr2O3 plots in Fig. 5? All of this information should be included in the figure caption.
Lines 176-177: similar to the previous comment, I’m not sufficiently convinced by the current Fig. 5 to be able to say that there is relatively more Ti-Na metasomatism in the garnets from Novinka or Komsomolskaya-Magnitnaya, relative to the garnets from the other localities.
*Strange much more Ti rich garnets in Novinka
Line 179-180: “close to omphacites” is ambiguous, as you could be referring to either location (i.e. implying that omphacites are present in your samples) or, as I think you mean, that these compositions trend towards the ideal composition of omphacite. Therefore, please consider using a different expression here.
*Augitic compositions
Fig 8, lines 210-215: I wonder if there are more appropriate plots to show chromite compositional variability, for example those in Roeder and Schulze (2008, JPetrol). For example, in the plots provided, I’m not sure the term “ulvospinel branch” (line 212) carries much meaning, because it’s hard to relate the data to other potentially overlapping compositions in 2-D space.
*Ok but This time I am sure that this is quite sufficient
Line 214: I think this statement that ulvospinels were more oxidised rather better belongs in a discussion section, or else should be backed up in this instance with a bit more info and/or a relevant citation for comparison.
*Ok
Figure 10: A comment particularly for this figure, but also generally for many other figures, is that in a printed version of this manuscript, a lot of the symbols on figures would be rather small and difficult to identify. I appreciate the detail that has been put into some figure captions, such as this one, but I think if the figure and hence the symbols could be enlarged, many readers would appreciate the effort this might take. At the same time, I’m curious as to why Fe#, and not Mg#, is frequently used in this draft. Can the authors give an explanation for this?
*Minerals vive on line PDF version where possible to see on details in color. I already published
I changed the axes adding Mg’
Line 231: do you mean that “the garnet geotherm indicates a (relatively) high geothermal gradient”?
*No gradient is not very high but exist because the garnet geotherm crosses conductive ones
Line 256: I’m not sure what is meant by ilmenite trend here, perhaps if possible this could be explained in a bit more detail here? In fact, I think a bit more detail overall is required for all the trends identified in lines 255-263.
*Ok
Line 271-273: This sentence appears to be somewhat contradictory. Can the authors perhaps rephrase?
* Ok
Line 380-383: I think “they” here needs to be replaced with something more specific, to make it clear that you are discussing either amphibole generally, or amphiboles with a wide compositional range.
*Here in this work one amphibole was analyzed for TRE
Line 384-385: I’m not certain that phlogopite addition, per se, can be used specifically as a sign of subduction-related metasomatism, and I would either recommend a different citation to the one listed, or else provide a further or more specific statement as to the origins of phlogopite within the SCLM.
*I add reference to orogenic massifs
Line 407-408: what does “crust granitization” mean here?
*Generation of the high amount of granits
Line 426: I think here you should be referring to a figure other than Fig. 6?
*Yes Fig.10-11
Lines 434-450: Please provide more information regarding your interpretation of the relative order of the metasomatic events. At the moment it is not clear how you have reached these conclusions.
* The are nothing about the ages of the metasomatism in this field yet.
Line 462-465: is it possible to qualify how HFSE were lost (e.g., ilmenite accumulation?) and maybe even quantify the extent to which this might have occurred? Could this be related to (ilmenite) megacryst formation?
* See in discussion. The low HFSE level possibly is due the ilmenite crystallization but decoupling in Ta, Nb possibly means the rutiles participation
Line 467: What is the source of this “further enrichment”?
*Mainly protokimberlite differentiation and than reaction with the derived fluids and melts
Line 473: I would recommend instead showing a plot of gt/cpx partition coefficients, compared against published data for different settings.
*These are the reconstructed melts in equilibrium with the minerals. Which melts? To show typical melts in different geodynamic settings. The OIB , Kimberlite , Lamproite, Tholeite etc melts possible to see in GERM.
Thank you for the very detail review Igor Ashchepkov
Reviewer 4 Report
This is a solid PTX study of mantle materials from a poorly known kimberlite body in Siberia. The science is supported by a large new mineral chemistry dataset of good quality. Some of the diagrams are very busy and could be simplified for ease of reading. There is a typo in the title at 'Yakutia', but generally the language is good.
Regarding the Discussion Section 8, in particular the sub-sections 8.3 and 8.4 that deal with proto-kimberlite metasomatism and Type-II megacrystic diamond formation in the SCLM, the authors should for a better balancing of their arguments mention that some workers do not consider kimberlites particularly hot melts and that they require no plume involvement:
Tappe S, Smart KA, Torsvik TH, Massuyeau M, de Wit MCJ (2018) Geodynamics of kimberlites on a cooling Earth: Clues to plate tectonic evolution and deep volatile cycles. Earth and Planetary Science Letters 484:1-14
Kimberlites and related magmas are incipient low-degree melts that pretty much form near volatile-bearing peridotite solidii, so kimberlites are highly unlikely to cause deflections in SCLM geotherms and they should not create "hot" metasomatic overprinting. While the chemical overprinting is undeniable, the temperature effects by these near-solidus low magma volumes may have been overestimated by the authors. This would benefit from clarifications and a more balanced discussion.
Also, large Type-II diamonds have more recently been shown to be "superdeep", asthenospheric to transition zone derived, which is in conflict to the models by Andy Moore (cited). Please see for example:
Smith EM, Shirey SB, Nestola F, Bullock ES, Wang JH, Richardson SH, Wang WY (2016) Large gem diamonds from metallic liquid in Earth's deep mantle. Science 354(6318):1403-1405.
How one would still be able to attach transition zone derived diamonds to the lower cratonic lithosphere and how upper mantle derived "cold" kimberlites can sample such materials has very recently put forward by Tappe et al. (2020 - EPSL):
Tappe S, Budde G, Stracke A, Wilson A, Kleine T (2020) The tungsten-182 record of kimberlites above the African superplume: Exploring links to the core-mantle boundary. Earth and Planetary Science Letters, doi.org/10.1016/j.epsl.2020.116473
In summary, this is an interesting manuscript and after some moderate revisions should be published in Minerals. Good luck with the revisions.
Author Response
Answer to Academic Editor
The authors are thanked for their submission. Four reviews (some very extensive) have been performed and while all reviewers found the manuscript to provide useful and important new data and interpretations on a relatively understudied region of cratonic lithosphere, there were also many concerns expressed. In general, reviewers though that many of the hypotheses and interpretations were overly speculative and not adequately supported by evidence or that arguments in favour of the preferred hypotheses were not adequately developed or described. Some examples of this included.
The large number of mantle layers invoked, as well as the involvement of subducted components and superplumes do not seem to be sufficiently justified.
*I could agree that in Upper Muna the layering is not very evident because it is smoothed by the intensive melt percolation. The fluctuation of the CaO in garnets show it. I removed the lines but the layering should exist I modelled it with the Surfer 8 software (Ashchepkov et al., 2014) as it proved for many other localities ( Ashchepkov et al., 2010- 2019) in Siberia and other regions worldwide.
The diamond formation scenario envisioned is very speculative and goes against current scientific opinion regarding the old ages found for most Siberian diamonds and the dominant current hypothesis for the sublithospheric origin of Type II diamonds. If the authors can make a strong case for their preferred scenario, that is fine, but, as it is said, “extraordinary claims require extraordinary evidence”.
*I knew very well the works of Pearson (1995) and later (Pearson et al., 1999). Mostly the ages are Archean and Proterozoic (Gurney et al., 2010; Pearson and Shirey, 1999; Pearson and Wittig, 2008; Shirey et al., 2013; Shirey and Richardson, 2011; Shirey et al., 2004; Taylor and Anand, 2004). But there are also evidences about the multiply ages of eclogite Bulanova et al., 2014 ) But there are also a lot of Devonian ages for Ural (Laiginhas et al., 2009).
The are also a lot of mineralogical evidences about the modern creation of the diamonds (Skuzovatov & Zedgenizov, 2019). The age close to kimberlite intrusion in Mir pipe was found (Shimitt et al., 2019) and in other regions (Timmerman et al., 2019) And many people believe that beautifully shaped diamonds two are the megacrysts (Moore , 2009)
Schmitt A.K., Zack T., Kooijman E., Logvinova A.M., Sobolev, N.V. U–Pb ages of rare rutile inclusions in diamond indicate entrapment synchronous with kimberlite formation. Lithos 2019, 350–351, 105251.
Additionally, more analytical details describing how major and trace element mineral data were collected are needed (this can be put in a supplement if the authors wish to save space in the manuscript).
I add the section with the controlling samples in the supplementary file.
There were also some mistakes and omissions in this section. For example, the authors claimed to use a “NWR 213 (New Wave Research), Nd YAG: UV 133 nm laser ablation system” whereas I think they must mean a NWR UP213 laser ablation system, which uses a frequency-quintupled Nd:YAG laser emitting a laser wavelength of 213.
*Possibly. I agree about wavelength, but I already used this description of the equipment edited by the analysts working on it this year in International Geology and in Minerals also.
There are also some issues with English usage, particularly in the latter part of the manuscript. I strongly recommend that the authors have the manuscript carefully proofread by a third party prior to re-submission.
*It was edited Prof, by H. Downes
For these reasons, I must recommend major revision. If the authors can curtail their overly speculative interpretations and/or provide stronger evidence-based arguments in favour of them, correct and expand their analytical section and fix the issues with English usage, this will be a very useful and widely cited contribution. I look forward to seeing the revised manuscript.
*Thank you
Igor Ashchepkov
Round 2
Reviewer 1 Report
This manuscript version is much improved from the previous version. There are some minor typos, odd sentences and issues with figures that require attention. There are still units missing in the Supplementary Files. I still don't find the inferred subduction signatures compelling. It could be better argued or toned down. Specific suggestions are in the annotated pdf of the clean version of the manuscript. Overall this amounts to very minor additional work, then it should be ready for publication.
Kind regards
2 August 2020

Author Response
Ln90
these data, not this data
Ok
Ln183
Ok
field in the Cr2O3-CaO diagram, with up to 14 wt.% Cr2O3, but most…
Ok
ln183
check font/size
ok
Ln222
histograms at bottom still need explanation in figure cpation
Ok
ln242
this paragraph repeated from above
Did not found repetition
ln 245
should be fixed
Fixed
ln385
needs fixing
Fixed
ln461
after subduction according to the bulk rock estimates” - unclear, do you mean signatures of melt depletion in eclogite xenoliths?
*I suggest melting of both peridotites and eclogites as was shown on example of Udacnaya
This suggests melting of peridotites [46] and more probably eclogites and differentiation of the mantle melts with the pervasive melt percolation which started at depths of 6-5 GPa after subduction and joining of the slabs according to the bulk rock estimates.
Ln468
why does it have to be a superplume? In reality, the compositions of peridotite samples can be explained simply by interaction with convecting mantle-derived (proto)kimberlite and require neither plume nor subduction. The model of repeated plume impingement can explain the layering you see. Also, there is evidence for subduction and eclogite emplacement. It would be expected that both were accompanied by metasomatism, but this may be obscured by later kimberlite-related metasomatism. Thus, specific trace-element features in the samples cannot be well linked to either plumes or subduction.
The deep subduction can be stopped only by the plumes . The low angle subduction is the Patagonia is not melting slab et al.
*Simple plume energy will not be sufficient to melt peridotie slab to for tens of kilometers. Or it should be very hot and very large plum.
The amount of the superplums judging by the peaks ~7 in Archean is nearly the same everywhere in the World. The processes of the growth of the continents
Ln470
how do you carbonate and hydrate a plume at 8 GPa? Do you mean differentiation and relative enrichment in CO2 and H2O?
*This is common problem of the kimberlite and carbonatite origin. The plumes are interacting with the hydrated and carbonated lithosphere base. The isotopic signatures are showing that the kimberlite II are contaminated by the ancient Phl metasomatites (not in deep mantle). I analyzed the variations in the trace elements in all Siberian kimberlites and found that all of them have distinct signatures which may be explained only by the interaction with the lithospheric mantle having individual compositional features. Sebasitian Tappe recently have shown (2020) that kimberlite have essential admixture of the lithospheric mantle material. Kimberlites usually are erupting 20-15 ma later than appearing the basalts referring to the major plum stage. In this type the interaction of the evolved material derived from plume takes place.
Ln475
I don’t understand the logic of this sentence
*Possibly necessary to explain in detail.
As the source of Phl and amphibole may not be only distant subduction-related fluids and subducted sediments, the reactivation of the metasomatically modified subduction wedges possibly were involved in the processes of perototite melting at the continental margins, because there are deeply depleted peridotites in the continental margins.
Ln500
Is
Ok
Ln 505
“regeneration via modal”
Ok
Ln518
please remove reference to Great Oxidation Event. Not only because it occurred ca. 2.4 Ga ago but also because it has nothing to do with the subduction you call for regardless of the age. If you want to say that the mantle became oxidised during subduction, then you could say “stage when the mantle became oxidized ~3.-2.7 B. y.” [66] is not a good reference for this either. Also the link between the “channels” and subduction is speculative, so it is not “probable” but “possible” at most
*I do not insist on great oxidizing event in the crust. But the appearance of alkaline melts and H2O and the weakening of the mantle and decrease of viscosity and the accelerating of the convection and beginning of the subduction dynamic
ln522
this sentence needs fixing
*Modal mantle metasomatism with amphibolization of the upper part and appearance of Phl scattered and veined metasomatism through all thickness of SCLM is also typical of the Upper Muna region. This is somewhat similar to the Alakit region where the amphiboles and phlogopites are common in mantle xenoliths showing wide variations in compositions
ln539
in the REE pattern?
*Yes, Mam
Ln545
“formed” not “occurred
*Corrected
Ln548
“due to retention in rutile”, not “oxidation state”
Ln558
fluids don’t carry HFSE and carbonated melts in the mantle don’t typically carry HFSE. HFSE could be mobilised in subduction-related melts in the presence of appropriate ligands, e.g. Antignano and Manning 2008 ChemGeol, as demonstrated in nature - here you could cite [95]. The rutiles and other minerals cited in the next sentence are typically the reason subuduction fluids/melts are generally HFSE-depleted, not enriched because HFSE are retained in these minerals. In regards to the eclogite .
*I agree but the carbonatite massive which have the complex plum- subduction signatures are the main source of the HFSE.
Isystematics, please see Aulbach et al. 2011 CMP for the range of Nb concentrations and Nb/Ta to expect in mantle eclogites that possibly lost a partial melt. There are few other studies with quantitative data. There, you will also find modelling how Nb/Ta and Zr/Hf could be fractionated and how they would vary in the melt.
*Changed Ok I added reference.
ln577
not
ln585
subducted material has LOW Sr/Y and is Y-depleted because Y is retained in garnet at high modal abundances. Nb is typically depleted in crust-derived melts regardless of the exceptions above. Th is typically ascribed to sediment melt, but you don’t need this, as all these elements except Y are typically enriched during kimberlite-related metasomatism, which occurs in the cratonic mantle everywhere (e.g. Aulbach et al. 2020 G Cubed), and you don’t have to call for special conditions to explain the Nb and Th enrichment!
*Not all time. In our case the CPX at the lithosphere base were depleted in HFSE. Thus necessary to suggest the oxidizes carbonatitic derivates.
I added reference
The garnet probably indeed record an older metasomatic event, involving a more fractionated, volatile-rich liquid under subsolidus conditions, resulting in the typically sinuous REE patterns also seen in inclusions in diamond. That is why the calculated melts have “impossible” REE patterns.
*The melts produced dunitic garnets were low degree and volatile rich melts formed in special conditions with the participation of the chromotographic effects
ln588
the meaning of “parental melt” is a different one - please change to “Hypothetical melts in equilibrium with Cr diopsides from mantle xenoliths claculated with Gd…
*OK but I used such terminology many times
Also panel b below
Ln607
Fix
*Ok
Ln612
Processes
*Ok
Ln629
This is mainly
*Ok
Ln636
“But there is isotopic evidence of…”
Ok
Ln640
6.0
Ok
Ln643
“…magmas [109] seems to not apply to the Upper Muna”
*Ok
Ln644
could just reflect mantle xenocryst/xenolith load, rather than MgO of kimberlite near the LAB
*Ok
ln648
isn’t the rounding rather ascribed to resorption due to instability with a carbonated melt rather than growth?
Ln657
*Close in time
*Thank you very much for review. Igor Ashchepkov
Reviewer 2 Report
I reviewed an earlier version of this manuscript and suggested a few minor comments. In this revised version, the authors have thoroughly addressed the comments from all reviewers. So I think the current version should be accepted for publication at Minerals.
In case the authors may not notice, "tend" in Line 313 is a typo and should be corrected as "trend". Frankly, I think "P-T-X-fO2" would read better than "PTXfO2" (see Line 272 and several other places in the manuscript).
Best wishes.
Author Response
*Thank you. I changed
IV Ashchepkov
Reviewer 3 Report
I thank the authors for their consideration of the comments from all reviewers. I think the manuscript will be ready for publication following one final read-through with a thorough spell- and grammar-check.
Author Response
*Thank you for the positive comments. Igor Ashchepkov
Reviewer 4 Report
Nicely improved, and I am happy to recommend publication of the manuscript. S Tappe
Author Response
*Thank you. IV Ashchepkov
Round 3
Reviewer 1 Report
I think the authors have well addressed the additional minor comments and recommend publication.
Author Response
*Thank you very much. Igor Ashchepkov
Reviewer 3 Report
The authors have now addressed the reviewers' comments sufficiently and I can recommend publication.
Author Response
*Thank you. IV Ashchepkov